# PaceLLM: Brain-Inspired Large Language Models for Long-Context Understanding

**Kangcong Li**[1*]**, Peng Ye**[2,3]*,**Chongjun Tu**[1]**, Lin Zhang**[1]**, Chunfeng Song**[2]**,
Jiamin Wu**[2]**, Tao Yang**[1]**, Qihao Zheng**[2†]**, Tao Chen**[1†]

[1]School of Information Science and Technology, Fudan University
[2]Shanghai Artificial Intelligence Laboratory
[3]The Chinese University of Hong Kong

## Abstract

While Large Language Models (LLMs) demonstrate strong performance across domains, their long-context capabilities are limited by transient neural activations causing information decay and unstructured feed-forward network (FFN) weights leading to semantic fragmentation. Inspired by the brain's working memory and cortical modularity, we propose PaceLLM, featuring two innovations: (1) a Persistent Activity (PA) Mechanism that mimics prefrontal cortex (PFC) neurons' persistent firing by introducing an activation-level memory bank to dynamically retrieve, reuse, and update critical FFN states, addressing contextual decay; and (2) Cortical Expert (CE) Clustering that emulates task-adaptive neural specialization to reorganize FFN weights into semantic modules, establishing cross-token dependencies and mitigating fragmentation. Extensive evaluations show that PaceLLM achieves 6% improvement on LongBench's Multi-document QA and 12.5–17.5% performance gains on $\infty$-Bench tasks, while extending measurable context length to 200K tokens in Needle-In-A-Haystack (NIAH) tests. This work pioneers brain-inspired LLM optimization and is complementary to other works. Besides, it can be generalized to any model and enhance their long-context performance and interpretability without structural overhauls.

## 1 Introduction

Large Language Models (LLMs) have revolutionized natural language processing, achieving state-of-the-art results in tasks ranging from open-ended text generation [3] to complex multi-step reasoning [47]. These advances have made LLMs the backbone of many real-world applications [43, 14, 41, 44, 42], from dialogue systems [24] to knowledge-intensive tasks [19]. As these applications scale, there is a growing demand for models to handle longer input sequences, particularly in scenarios such as multi-document question answering [50], long-form summarization [30], and conversational memory [36]. Modeling such extended contexts requires LLMs not only to retain information over longer spans, but also to reason over distributed and interdependent content. This has brought renewed attention to the internal mechanisms that govern context modeling and memory persistence within LLMs.

Existing approaches to address long-context challenges generally fall into three categories. The first enhances LLMs' reasoning capacity through architectural or training improvements [27, 40, 37, 32]. The second focuses on input compression, reducing redundancy while preserving key information [23,

---

*Equal contribution. † Corresponding authors: `zhengqihao@pjlab.org.cn,eetchen@fudan.edu.cn`.

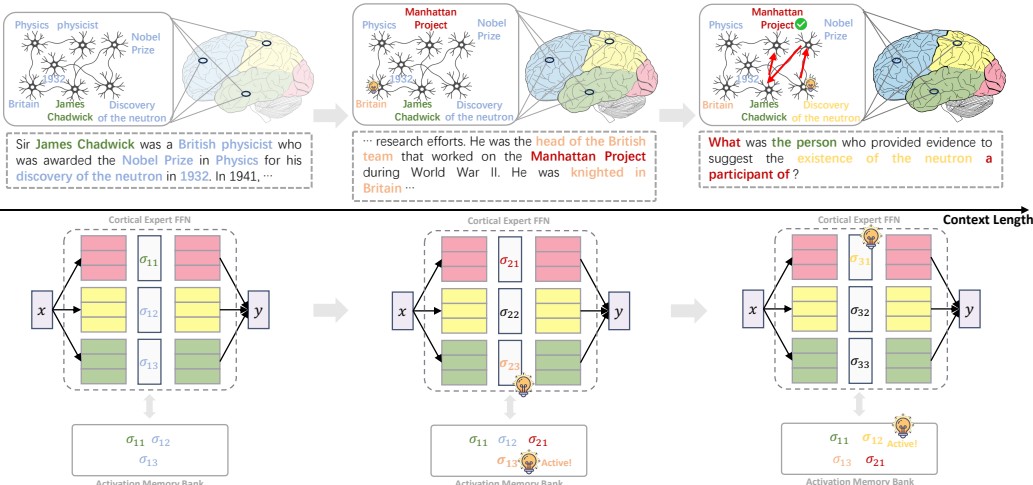

**Figure 1:** Schematic diagram of the PaceLLM (bottom) and its neuroscience counterpart (top). In this case, which introduces James Chadwick's character, the brain processes and retains key information through working memory. When the content in working memory appears in the subsequent text, such as "Britain", relevant neurons will persistently to be re-active. When the final question is input, the neuron with the keyword "neutron" will also persist to be re-activated, connect with other relevant neurons, and finally find the answer "Manhattan Project". Analogical to the mechanism of brain, PaceLLM expertly clustered FFN weights, and designed an Activation Memory Bank (AMB) to interact with activations.

26, 31, 45, 9]. The third introduces external components, such as memory modules [34, 7] and retrieval-augmented generation (RAG) [49, 33, 38], to compensate for limited attention spans. However, these approaches often overlook a fundamental internal limitation: the role of feed-forward networks (FFNs). Specifically, transient neural activations cause information to fade over time, and unstructured FFN weights may fragment semantics across tokens, jointly undermining coherence in long context understanding.

To alleviate this problem, we draw inspiration from neuroscience to explore the untapped potential of FFN activations. Notably, the brain's working memory [51] and cortical modularity [1] demonstrate remarkably effective mechanisms for long-context processing, as illustrated at the top of Figure 1. Working memory refers to the brain's ability to temporarily retain and manipulate task-relevant information through persistent neural activity in the prefrontal cortex (PFC) [8]. When previously stored information reappears, relevant PFC neurons remain active, helping preserve relevant content and counteract information decay. Concurrently, the cerebral cortex [25] is functionally partitioned into distinct regions [15], enabling specialized "neuron experts" to handle different tasks. This modular organization improves semantic consistency and supports efficient long-context comprehension.

Inspired by above brain's mechanisms, we propose PaceLLM, as illustrated at the bottom of Figure 1. Our approach consists of two key components: (1) Activation Memory Bank (AMB) to emulate PFC persistent activity (PA). This component flattens, retrieves, fuses, and stores intermediate activations. Retrieval computes similarity between current and historical activations, allowing highly similar representations to be reactivated and reused. (2) Cortical Expert (CE) via clustering and reordering. We first cluster the gated projection matrix with equal experts per cluster. Then, the gated and upper projection matrices are reordered by rows, and the lower projection matrix is reordered by columns, yielding a structured FFN with expert-specialized layout.

We evaluate the proposed PaceLLM on LongBench [2] and $\infty$-Bench [46] using Qwen-2-7B-Instruct [39] and Llama-2-7B-chat [28] as base models. Under the training-free setting, our method consistently outperforms baselines. When aligned with fine-tuning baselines, we achieve a 6% improvement on the Multi-document QA task in LongBench. On $\infty$-Bench, the performance of En.Dialogue and En.Multi-Choice tasks is improved by 12.5% and 17.5%, respectively. In the Needle-In-A-Haystack (NIAH)[18] test, our method handles contexts up to 200K tokens, substan-

tially surpassing Activation Beacon[45]'s 128K limit. Our contributions can be summarized as follows:

**(1) A pioneering brain-inspired approach to enhance LLMs' long-context understanding.** While prior efforts achieve great success, they overlook internal inefficiencies—specifically, fleeting activations that weaken retention and disordered FFN weights that disrupt semantic continuity. We propose the first brain-inspired solution targeting these core limitations.

**(2) Training-free persistent activity (PA) and cortical expert (CE) clustering mechanisms.** We introduce a memory bank that mimics working memory by operating at the activation level, enabling finer-grained retention than token-level storage. Our cortical modularity method structures FFNs to better capture inter-token semantics. Our method is model-agnostic and plug-and-play.

**(3) Strong performance across long-context benchmarks and NIAH.** Our approach achieves over 10% gains on several tasks and extends the usable context length to 200K tokens, demonstrating both improved reasoning capabilities and robust scalability.

## 2 Related Work

### 2.1 Modeling and Understanding Long Contexts with LLMs

Enhancing LLMs' ability to process long contexts remains an active research challenge with three mainstream directions. Input preprocessing techniques like prompt engineering [27, 48], position encoding [26, 6] and KV cache compression [22, 31, 45] reduce input complexity and guide LLMs to focus on key information; LLM structural optimizations, such as continual learning [37] and model editing [32], adapt model parameters to better handle extended contexts. External augmentation methods, including memory banks [34, 7] and Retrieval-Augmented Generation [49, 33, 38], supplement the model's internal capabilities by storing historical information or retrieving relevant content. Despite demonstrated improvements, these approaches have limitations: preprocessing methods often operate at coarse granularity (token or embedding level), structural optimizations incur significant computational costs, and external augmentations introduce system complexity and operational overhead.

It has been increasingly recognized that feed-forward networks (FFNs) in Transformers operate as key-value memories, where each neuron responds to specific input patterns and produces associated outputs [10]. Our proposed PaceLLM differs from existing studies by focusing on the feed-forward networks (FFNs) within transformer layers, an aspect largely overlooked in previous long-context solutions. PaceLLM addresses two core issues: transient neural activations causing information decay and unstructured FFN weights leading to semantic fragmentation. Our approach operates at activation-level granularity and reorganizes FFN weights into semantic modules, providing a complementary solution that can be integrated with existing methods to further enhance long-context understanding.

### 2.2 Brain-Inspired Interpretability in LLMs

Brain-inspired approaches have emerged as a promising direction for improving LLM interpretability and performance. HippoRAG [17] implements a retrieval system modeled after neocortex-hippocampus interactions. HMT [11] introduces a three-level memory hierarchy mimicking human memory processes. Larimar [5] augments LLMs with an external episodic memory module for knowledge editing and long-context processing. NeuroMFA [35] quantifies emergent abilities in LLMs by analyzing structural dynamics of neuron interaction networks. These approaches demonstrate how mechanisms in the brain can enhance model architecture, processing mechanisms, and interpretability, establishing valuable cross-disciplinary connections.

PaceLLM extends brain-inspired research by focusing on neural persistent activity (PA) and cortical expert (CE), which are two underexplored yet fundamental neurobiological principles. In contrast to prior work emphasizing external modules or attention layers, our method targets the FFNs, which account for most model parameters but lack neuroscience-guided design. By embedding activation-level memory and expert clustering into the computation flow, PaceLLM enhances long-context performance with minimal architectural changes.

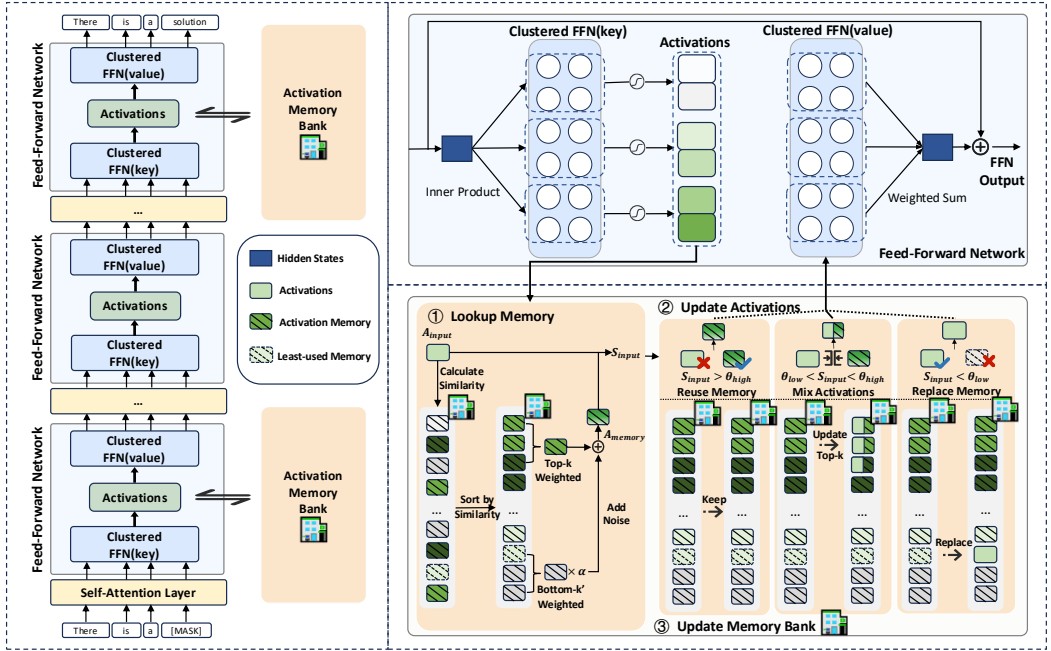

**Figure 2:** The illustration of PaceLLM. The left of the figure is an overall pipeline. Note that Activation Memory Bank (AMB) doesn't interact with all FFN layers. The top right of the figure is a detailed illustration of the modified FFN layer. The bottom right is a detailed processing flow of AMB. ①Lookup Memory shows the process of similarity retrieval, taking the top$k$, and adding noise. ② shows the selection of reusing strategies by comparing similarity with threshold. ③ shows three strategies for updating the AMB.

## 3 Method

### 3.1 Preliminary

Modern LLMs are primarily built upon the Transformer [29] architecture, which contains two core components: the multi-head self-attention mechanism and the position-wise feed-forward network (FFN). While attention modules enable dynamic global interactions, FFNs process token-level information in parallel and contribute substantially to the model's capacity and computational cost.

**Multi-Head Self-Attention.** It dynamically models global contextual dependencies between tokens by computing attention scores across all positions in the sequence:

$$
\begin{aligned}
\mathbf{Attention}(Q,K,V) &= \mathbf{softmax}\left(\frac{QK^\top}{\sqrt{d_k}}\right)V \\
\mathbf{MultiHead}(Q,K,V) &= \mathbf{Concat}(\mathbf{head}_1,\ldots,\mathbf{head}_h)W^O
\end{aligned}
\tag{1}
$$

where $\mathbf{head_i} = \mathbf{Attention}(QW_i^Q, KW_i^K, VW_i^V)$.

**Position-wise Feed-Forward Network.** It applies non-linear transformations to refine individual token representations, operating independently on each position. For an input token representation $\mathbf{x} \in \mathbb{R}^{d_{\text{model}}}$, the FFN layer performs:

$$
\mathrm{FFN}(\mathbf{x}) = \mathbf{W}_2 \cdot \sigma(\mathbf{W}_1\mathbf{x} + \mathbf{b}_1) + \mathbf{b}_2
\tag{2}
$$

where $\mathbf{W}_1 \in \mathbb{R}^{d_{\text{ff}} \times d_{\text{model}}}$ and $\mathbf{W}_2 \in \mathbb{R}^{d_{\text{model}} \times d_{\text{ff}}}$ are learnable weights. $d_{\text{ff}}$ typically set as $4d_{\text{model}}$ defines the expanded intermediate dimension. Activation function $\sigma$ (e.g., ReLU, GeLU) enables non-linear feature interactions.

## 3.2 PaceLLM

Inspired by working memory and cortical processing in the brain, we propose **PaceLLM (Persistent Activity and Cortical Experts LLM)** to enhance long-context understanding. As shown in Figure 2, PaceLLM integrates two biologically motivated components: (1) *Activation Memory Bank (AMB)*, which mimics persistent neural activity in working memory by caching and retrieving FFN activations; and (2) *Cortical Experts (CE) Clustering*, which introduces a similarity-based expert selection mechanism, inspired by specialized processing in the cerebral cortex. We describe each component below.

### 3.2.1 Activation Memory Bank (AMB)

To simulate persistent neural activity, we augment the FFN with an Activation Memory Bank (AMB) that stores and reuses intermediate activations. Specific FFN layers are equipped with a memory bank $\mathcal{M} = \{\mathbf{K}, \mathbf{V}, \mathbf{u}\}$ where $\mathbf{K}, \mathbf{V} \in \mathbb{R}^{M \times d_{\text{ff}}}$ denote the memory keys and values, and $\mathbf{u} \in \mathbb{R}^M$ tracks usage frequency. The workflow consists of memory lookup, activation update, and memory update.

**Memory Lookup.** Given intermediate activations $\mathbf{X}_c \in \mathbb{R}^{C \times d_{\text{ff}}}$, we compute their cosine similarity with stored keys $\mathbf{K}$:

$$\text{sim}_{ij} = \frac{\mathbf{x}_i^\top \mathbf{k}_j}{\|\mathbf{x}_i\|\|\mathbf{k}_j\|}, \quad \forall i \in [C], j \in [M]. \tag{3}$$

We then retrieve the top-$k$ most similar historical entries and bottom-$k'$ least similar ones to introduce diversity:

$$\mathbf{I}^{\text{top}}, \mathbf{S}^{\text{top}} = \text{TopK}(\text{sim}, k), \tag{4}$$

$$\mathbf{I}^{\text{neg}}, \mathbf{S}^{\text{neg}} = \text{TopK}(-\text{sim}, k'). \tag{5}$$

**Activation Update.** The final output $\mathbf{o}_i$ is computed by integrating current and retrieved activations based on similarity confidence:

$$\mathbf{o}_i = \begin{cases} \boldsymbol{\mu}_i^{\text{pos}} + \lambda \boldsymbol{\mu}_i^{\text{neg}}, & \text{if } \max(\mathbf{S}_i^{\text{top}}) > \theta_{\text{high}} \\ \text{Avg}(\mathbf{x}_i, \boldsymbol{\mu}_i^{\text{pos}}) + \lambda \boldsymbol{\mu}_i^{\text{neg}}, & \theta_{\text{low}} < \max(\mathbf{S}_i^{\text{top}}) \leq \theta_{\text{high}} \\ \mathbf{x}_i, & \text{otherwise} \end{cases} \tag{6}$$

where $\boldsymbol{\mu}_i^{\text{pos}}$ and $\boldsymbol{\mu}_i^{\text{neg}}$ are mean vectors of top and bottom activations, and $\lambda$ is a noise scaling factor.

**Memory Update.** After computing outputs, AMB is updated using a similarity-aware strategy:

- High similarity ($\overline{\mathbf{S}^{\text{top}}} > \theta_{\text{high}}$): No update; only increment usage counter $\mathbf{u}$.
- Medium similarity ($\theta_{\text{low}} < \overline{\mathbf{S}^{\text{top}}} \leq \theta_{\text{high}}$): Update stored memory by merging current activation:

$$\mathbf{K}_j \leftarrow \text{Avg}(\mathbf{K}_j, \boldsymbol{\mu}_c), \quad \mathbf{V}_j \leftarrow \text{Avg}(\mathbf{V}_j, \boldsymbol{\mu}_c) \tag{7}$$

- Low similarity ($\overline{\mathbf{S}^{\text{top}}} \leq \theta_{\text{low}}$): Replace least-used slot using LRU policy [4].

While CAMELoT [12] uses similarity to trigger memory updates based on novelty, it replaces the least recently used slot, ignoring semantic importance. In contrast, PaceLLM selectively retains and updates memory based on both similarity and contextual relevance, mimicking persistent neural activity in working memory. This mechanism allows PaceLLM to persist and reuse relevant activation traces dynamically across long contexts.

### 3.2.2 Cortical Expert (CE) Neuron Clustering

Inspired by the functional modularity of the brain cortex [1], where localized neuron groups are activated by similar input signals, we reinterpret the FFN layer as an overparameterized neuron pool that can be decomposed into semantically coherent *cortical experts*. This decomposition enables both specialization and modularity in later decoding. We propose a two-stage transformation of pretrained FFN weights: (1) expert discovery via balanced clustering, and (2) parameter reorganization to form modular expert blocks. This design mirrors cortical specialization, where neurons with similar activation properties co-locate and collaborate. This process does not require retraining.

**Table 1:** Performance comparison between PaceLLM and baseline models on LongBench tasks in **training-free** setting. CE denotes cortical expert neuron clustering and PA denotes persistent activity memory mechanism.

| Model | Method | SQA | MQA | Sum. | FSL | Cod. |
|---|---|---|---|---|---|---|
| Qwen-2-7B-Instruct | Vanilla | 37.76 | 49.03 | 28.93 | 70.36 | 50.05 |
| | Vanilla + CE | 37.68 | 48.80 | 28.85 | 70.61 | **50.36** |
| | Vanilla + PA | 38.09 | 49.36 | 28.86 | 70.92 | 49.60 |
| | Vanilla + CE + PA | **38.49** | **50.28** | **29.02** | **70.96** | 49.95 |
| Llama-2-7B-chat | Vanilla | 23.92 | 23.42 | 24.43 | 63.02 | **55.48** |
| | Vanilla + CE | 24.49 | 23.73 | 24.38 | 62.86 | 55.17 |
| | Vanilla + PA | 24.65 | 23.15 | 24.18 | 63.23 | 54.98 |
| | Vanilla + CE + PA | **25.35** | **23.75** | **24.61** | **63.58** | 55.28 |

**Expert Discovery via Constrained Clustering.** Given FFN weight matrices $\mathbf{W}_1 \in \mathbb{R}^{d_{\text{ff}} \times d_{\text{model}}}$ and $\mathbf{W}_2 \in \mathbb{R}^{d_{\text{model}} \times d_{\text{ff}}}$, we treat the rows of $\mathbf{W}_1$ as candidate neurons and apply KMeansConstrained [21] clustering:

$$\tilde{\mathbf{w}}_i = \frac{\mathbf{w}_i}{\|\mathbf{w}_i\|}, \quad \forall i \in \{1, \ldots, d_{\text{ff}}\} \tag{8}$$

$$\min_{\{C_j\}} \quad \sum_{j=1}^{K} \sum_{i \in C_j} \|\tilde{\mathbf{w}}_i - \mu_j\|^2 \quad \text{s.t.} \quad |C_j| = \frac{d_{\text{ff}}}{K} \tag{9}$$

where $K$ is the predefined number of experts and $C_j$ denotes the cluster for expert $j$.

**Parameter Reorganization.** Let $\pi$ be the index permutation obtained by concatenating all cluster memberships. We reorganize FFN weights as follows:

$$\mathbf{W}_1^{\text{new}}[iK : (i+1)K, :] = \mathbf{W}_1[\pi[iK : (i+1)K], :] \tag{10}$$

$$\mathbf{W}_2^{\text{new}}[:, iK : (i+1)K] = \mathbf{W}_2[:, \pi[iK : (i+1)K]] \tag{11}$$

This expert-wise rearrangement preserves the integrity of each neuron cluster while maintaining compatibility with the original FFN structure.

**Implementation Details.** Caching: Expert indices are cached per layer to avoid redundant clustering during repeated runs. In-place Processing: Reordering is performed in-place to reduce memory overhead. Inference Compatibility: Output shapes and computational graphs remain unchanged, ensuring zero-cost integration.

## 4 Experiments

### 4.1 Settings

**Datasets.** We evaluate PaceLLM on three established long-context benchmarks: LongBench [2], $\infty$-Bench [46] and Needle-In-A-Haystack (NIAH) [18]. To evaluate the generalization ability of our method beyond long-context tasks, we also evaluate on MMLU [13], which features shorter context lengths.

**Implementation.** We apply PaceLLM to Llama-2-7B-chat [28] and Qwen-2-7B-Instruct [39] in training-free and low-cost fine-tuning settings. For low-cost fine-tuning, we follow the setting of Activation Beacon [45]. All experiments are conducted with 4×A100-40G GPUs.

**Baselines.** We compare PaceLLM with the original base models and several context compression methods, including LongLLMLingua [16], SnapKV [20], and Activation Beacon (AB) [45]. As PaceLLM is orthogonal to these methods, we also integrate PaceLLM with Activation Beacon to demonstrate complementary benefits.

### 4.2 Experimental Results

**Results on LongBench.** Table 1 presents training-free performance results. For both Qwen-2 and Llama-2, the components of our method (cortical expert neuron clustering CE and persistent activity

**Table 2:** Performance comparison between PaceLLM and baseline models on LongBench tasks in **low-cost fine-tuning** setting. CE denotes cortical expert neuron clustering, and PA denotes persistent activity memory mechanism.

| Model | Method | SQA | MQA | Sum. | FSL | Cod. |
|---|---|---|---|---|---|---|
| | Vanilla-FT | 41.00 | 40.60 | 26.80 | 68.50 | 66.10 |
| | LongLLML [16] | 24.70 | 20.30 | 26.30 | 55.90 | 50.10 |
| | SnapKV [20] | 38.70 | 37.60 | 26.20 | 67.10 | 60.30 |
| Qwen-2-7B-Instruct | Activation Beacon [45] | 40.50 | 40.30 | 26.80 | 68.40 | 66.40 |
| | Activation Beacon + PA | 41.10 | 42.80 | 27.90 | 69.31 | 67.51 |
| | Activation Beacon + CE | 40.90 | 44.58 | 27.36 | 68.98 | 67.26 |
| | Activation Beacon + CE + PA | **42.62** | **46.55** | **28.74** | **70.56** | **67.52** |

<table>
<tr><td colspan="7">Table 3: Results on ∞-Bench.</td></tr>
<tr><td></td><td>En.Dia</td><td>En.Sum</td><td>En.QA</td><td>Zh.QA</td><td>En.MC</td><td>Code.Run</td></tr>
<tr><td>AB [45]</td><td>3.00</td><td>3.37</td><td>9.57</td><td>22.34</td><td>46.72</td><td>0.50</td></tr>
<tr><td>Ours</td><td>15.5</td><td>4.11</td><td>14.14</td><td>24.84</td><td>64.19</td><td>2.50</td></tr>
</table>

**Table 4:** Results on MMLU.

| | STEM | Social Sciences | Humanities | Others | Avg. |
|---|---|---|---|---|---|
| AB [45] | 61.891 | 79.780 | 72.724 | 70.530 | 70.250 |
| Ours | **61.974** | **80.047** | **72.915** | **71.075** | **70.510** |

memory PA) individually improves the performance. When combined, they work synergistically and achieve the best overall performance, with improvement up to 1.4% on certain subtasks without any training. To ensure fairness compared to the fine-tuning method, Table 2 shows PaceLLM's low-cost fine-tuning performance. Applying our method to Activation Beacon [45] leads to significant performance improvements across all task categories, especially the Multi-document QA task having improved **6%** performance. The consistent performance gains demonstrate that our brain-inspired approach effectively enhances the model's ability to process long-range contextual information. The best performance is achieved by combining the two mechanisms.

**Results on ∞-Bench.** Table 3 shows the experimental results on ∞-Bench, another long-context benchmark. Without any additional training, our method outperforms Activation Beacon significantly across all tasks. For example, **12.5%** on En.Dialogue task and **17.5%** on En.Multi-Choice task.

**Results on Needle-In-A-Haystack.** We further evaluate on Needle-In-A-Haystack (NIAH) following the official settings [18] and illustrate the results in Figure 3. The context length is expanded to 200K for further evaluation. As can be concluded, our proposed PaceLLM consistently retrieves the needle more precisely than Activation Beacon's 128K context length.

**Results on MMLU.** As can be seen from Table 4, while our method is specially designed for long-context scenarios, it maintains performance improvements on the short-context MMLU benchmark. This indicates that PaceLLM has not compromised in its general language understanding capabilities.

## 4.3 Discussion

The experimental results of each model on different datasets can prove the effectiveness of PaceLLM. To further improve the interpretability of our method, we also design a visualization experiment. The selected model is Qwen2-7B and the task is GovReport in LongBench. As shown in the Figure 4, during model evaluation, we record activations from both current input and AMB at different moments and convert them back to tokens with semantics. According to the semantic information, they are drawn in a two-dimensional semantic figure, where points with similar distances indicate similar semantics, the color of the points indicates the usage frequency according to the legend on the right, and the red point indicates the activation corresponding to the current input.

The visualization shows that the current input activation form clusters with semantically similar historical activations, while the historical activations in each cluster are fully reused. Therefore, it can be inferred that PaceLLM can retrieve the semantically similar historical activations stored in AMB for different current activations, which can be re-activated and reused sufficiently many times by analogy with working memory. This demonstrates that PaceLLM indeed has a mechanism highly similar to the brain's working memory, which effectively enhances the understanding of long contexts.

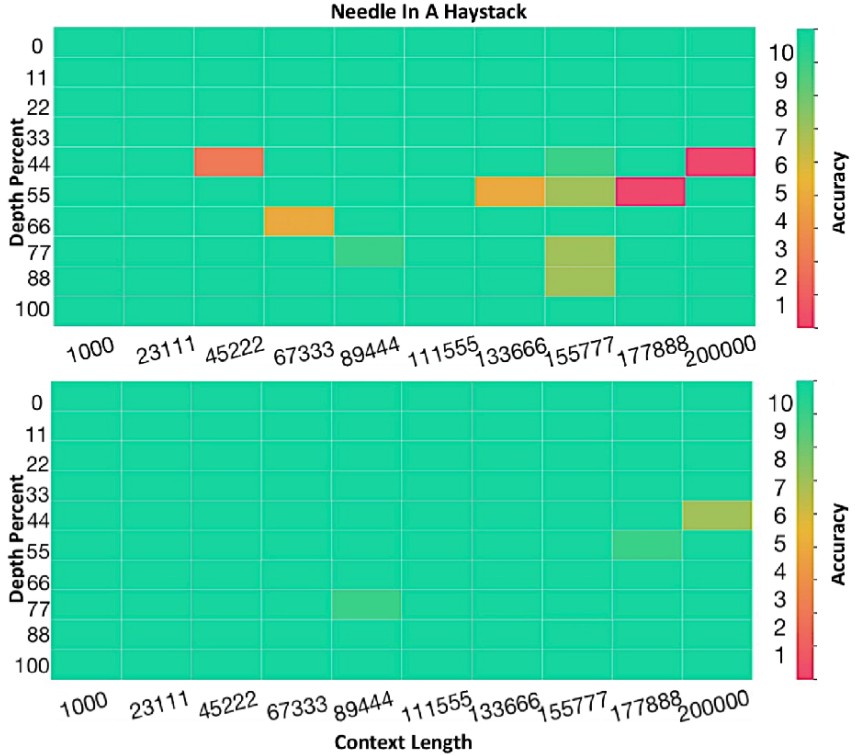

**Figure 3:** Evaluation on Needle-In-A-Haystack. PaceLLM (bottom) can retrieve the needle up to 200K than Activation Beacon 128K (top).

### 4.4 Ablation Studies

To facilitate a fair and systematic comparison, we establish a base configuration using Qwen2-7B on LongBench. In the base setting, the bank capacity $M$ is set to 100, the fusion threshold $\theta_{\text{high}}$ is 0.7 and $\theta_{\text{low}}$ is 0.3, with AMB applied to the 13[th] and 27[th] layers. Based on this setting, we conduct ablation studies about deployment location, fusion thresholds, and the design of noise adding in memory lookup as follows.

**Ablation of deployment location.** Since our approach is flexible and can be integrated into any layer of the model, we examine the effect of applying our method at different network depth and report the results in Table 5 (a). For single-document question answering (SQA) and code generation (Cod.), sparse deployment (e.g., layers 13 and 27) performs better due to lower requirements for long-range coherence and higher variability in input texts. For summarization (Sum.) and multi-document question answering (MQA), which demand stronger global context modeling, denser layer configurations (e.g., every other or fourth layer) yield better results. Deploying at all layers consistently underperforms and increases computational cost. Therefore, we adopt different sparse deployment locations for different tasks.

**Ablation of fusion thresholds.** Table 5 (b) shows the impact of different fusion thresholds $\theta_{\text{high}}$ and $\theta_{\text{low}}$ across tasks. For complex tasks such as MQA, Sum., and few-shot learning (FSL), better results are achieved with lower $\theta_{\text{low}}$ (e.g., 0.1), indicating that direct reuse of high-similarity activations from the AMB improves consistency and coherence in long-range context modeling. Among these, MQA particularly benefits from combining current and historical representations, suggesting its need for both contextual understanding and knowledge retrieval. In contrast, for simpler tasks like SQA or code generation, where input contexts are shorter and exhibit less inter-dependency, moderate thresholds (e.g., 0.5) yield optimal performance. This suggests that excessive memory reuse may introduce noise rather than useful information for such tasks.

**Ablation of noise adding design.** Results in Table 5 (c) confirm that adding negative entries (Equation 5) into activations consistently improves performance. This design draws inspiration

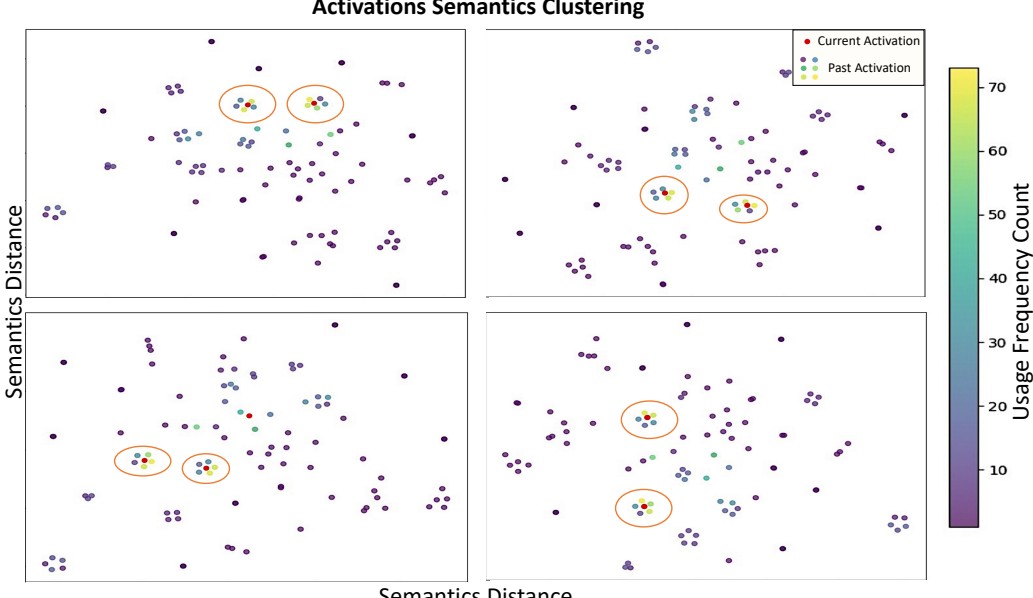

**Figure 4:** Visualization of current and historical activations. The orange circles encircled the clusters of current and past activations, which means they have similar information and useful past activations are sufficiently reused. It illustrates PaceLLM leverages the AMB to retrieve semantically similar past activations, enabling repeated reuse in a manner analogous to working memory.

**Table 5:** Performance comparison: (a) across different network layers, (b) under various fusion threshold settings, and (c) with/without noise addition.

(a) Applied at different network layers

| Layer No. | SQA | MQA | Sum. | FSL | Cod. |
|---|---|---|---|---|---|
| Baseline | 40.90 | 44.58 | 27.36 | 68.98 | 67.26 |
| 1,2,3,···,28 | 40.48 | 44.94 | 28.39 | 68.59 | 64.05 |
| 2,4,6,···,28 | 41.20 | **45.49** | **28.78** | 68.02 | 65.48 |
| 2,6,10,···,26 | 41.52 | 45.34 | 28.47 | 69.90 | 66.49 |
| 2,10,18,26 | 41.28 | 44.84 | 28.02 | **69.96** | 66.81 |
| 2,27 | 41.46 | 45.41 | 28.36 | 69.79 | **67.36** |
| 14,26 | 41.23 | 45.03 | 28.34 | 69.52 | 66.64 |
| 13,27 (base) | 41.62 | 44.68 | 28.19 | 69.41 | 66.71 |
| 1 | 41.53 | 45.12 | 28.38 | 69.35 | 66.97 |
| 26 | **41.67** | 44.97 | 28.17 | 69.16 | 66.98 |
| 14 | 41.21 | 44.77 | 28.60 | 69.33 | 67.09 |

(b) Under various fusion threshold settings

| $\theta_{\text{high}}, \theta_{\text{low}}$ | SQA | MQA | Sum. | FSL | Cod. |
|---|---|---|---|---|---|
| Baseline | 40.90 | 44.58 | 27.36 | 68.98 | **67.26** |
| 0.9, 0.9 | 40.97 | 44.98 | 28.44 | 69.09 | 66.98 |
| 0.1, 0.1 | 41.43 | 45.49 | **28.64** | **70.1** | 66.94 |
| 0.5, 0.5 | 41.58 | 45.13 | 28.56 | 68.83 | 67.13 |
| 0.9, 0.1 | 40.52 | **45.83** | 28.57 | 69.71 | 65.64 |
| 0.7, 0.3 (base) | **41.62** | 44.68 | 28.19 | 69.41 | 66.71 |

(c) Ablation results with and without noise addition

| Setting | SQA | MQA | Sum. | FSL | Cod. |
|---|---|---|---|---|---|
| with noise (base) | **41.62** | **44.68** | **28.19** | **69.41** | **66.71** |
| w/o noise | 40.70 | 43.90 | 27.90 | 68.97 | 66.49 |

from human memory systems, where both relevant and contrasting information contribute to robust decision-making. For each query, if all top-$k$ samples are extremely similar, introducing a small number of least-similar samples can serve multiple purposes, such as providing additional context or counter-examples, preventing excessive repetition, and enhancing adaptability to diverse scenarios.

## 5 Conclusions & Limitations

Inspired by the prefrontal cortex's working memory and cerebral modularity, we propose PaceLLM, a brain-inspired framework to enhance long-context understanding in LLMs. Our method introduces two key innovations: Persistent Activity Memory Mechanism (PA) dynamically retrieves and reuses FFN activations through an external Activation Memory Bank (AMB), simulating the persistent firing. By selectively storing high-value activations and employing similarity-based fusion strategies, this mechanism mitigates context degradation in long sequences. Cortical Expert Neuron Clustering (CE) reorganizes disordered FFN weights into task-specialized modules, establishing semantic links

between isolated token representations. This mimics the brain's cortical modularity. Experimental results demonstrate significant improvements across multiple benchmarks.

Our method has great highlights in performance, biological plausibility and interpretability of LLMs. It is the first brain-inspired improvement in the FFN layer for solving long-context problems, which is complementary to most existing methods and is plug-and-play. However, AMB is an additional module based on the original model, which will introduce certain extra calculation and storage costs. In addition, given that our method is orthogonal to most works, we believe that our method will not be limited to the field of plain text understanding, and we can extend our method to multi-modal, embodied intelligence and other fields in the future to fully realize the potential of brain-inspired AI technology progress.

## Acknowledgments and Disclosure of Funding

This work is supported by National Key Research and Development Program of China (No. 2022ZD0160101), Shanghai Natural Science Foundation (No. 23ZR1402900), Shanghai Science and Technology Commission Explorer Program Project (24TS1401300), Shanghai Municipal Science and Technology Major Project (No.2021SHZDZX0103) and Shanghai Artificial Intelligence Laboratory. The computations in this research were performed using the CFFF platform of Fudan University.

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

# A    Inference Efficiency Analysis

To quantitatively assess the computational overhead introduced by our proposed method PaceLLM, we conduct a series of rigorous inference time measurements on the Qwen2-7B model using the Qasper task from LongBench—a representative long-context question answering benchmark. Our evaluation focuses on both absolute inference time and relative time increase compared to baseline methods. The results are summarized in Table 6.

**Table 6:** Inference Time Comparison on Qwen2-7B (Qasper Task)

| Method | Sdpa Attention | Flash Attention |
|---|---|---|
| Vanilla | 7m31s | 7m03s |
| Activation Beacon | 4m42s | 4m19s |
| Ours | 6m32s | 6m09s |

**Controlled Time Overhead.**    Compared to the most efficient baseline (Activation Beacon), our method introduces a moderate and controlled increase in inference latency. Specifically, the relative time overhead is approximately ×1.37 with SDPA and ×1.32 with FlashAttention. However, compared to the Vanilla baseline without any memory mechanism, our method achieves a significant speedup—about 13.2% faster under SDPA and 13.4% faster under FlashAttention. This highlights that our approach strikes a favorable balance between computational complexity and memory-enhanced modeling capability.

**Compatibility with Attention Optimizations.**    All methods benefit from attention-level optimization. Transitioning from SDPA to FlashAttention yields a consistent 6–7% speedup across all setups. Importantly, our method is fully compatible with FlashAttention, demonstrating its practical applicability to real-world, performance-critical environments.

**Breakdown of Overhead Sources.**    The primary computational overhead in our method stems from the activation memory mechanism, including dynamic activation storage, similarity-based lookup, and selective activation reconstruction. These components are central to the model's ability to capture and reuse long-range dependencies. Nonetheless, they are designed to be lightweight, ensuring that the overall throughput remains practical.

**Efficiency–Performance Trade-off.**    The additional inference time is well justified by the performance gains observed in multiple long-context tasks. Compared to the Vanilla baseline, our method reduces latency while improving comprehension. Compared to the Activation Beacon, we achieve stronger results with acceptable overhead. For latency-sensitive applications, the design of our system offers a tunable trade-off between inference efficiency and accuracy.

**Summary.**    PaceLLM maintains operational feasibility with predictable computational cost. It integrates well with widely adopted acceleration techniques such as FlashAttention and provides a favorable performance–efficiency trade-off, making it suitable for both research and real-world deployment scenarios.

# B    Detailed Performance on LongBench

Table 7 reports the performance of PaceLLM on a variety of long-context understanding tasks from LongBench in a training-free setting. We evaluate two major foundation models—Qwen-2 and Llama-2—and progressively apply our brain-inspired mechanisms: cortical expert neuron clustering (CE) and persistent activity memory (PA).

**Component-wise Improvements.** Individually, both CE and PA contribute positively across most tasks. For Qwen-2, CE enhances performance particularly in *Single-Document QA* (e.g., NrtvQA improves from 25.38 to 25.87) and *Code* tasks (e.g., RB-P rises from 46.47 to 46.71). PA, on the other hand, is especially effective in *Few-shot Learning* (e.g., TREC from 76.00 to 78.00) and

**Table 7:** Performance comparison between PaceLLM and baseline models on LongBench tasks in **training-free** manner. CE denotes cortical expert neuron clustering and PA means persistent activity memory mechanism.

| | Method | Single-Document QA | | | | Multi-Document QA | | | | Summarization | | | | Few-shot Learning | | | | Code | | |
|---|---|---|---|---|---|---|---|---|---|---|---|---|---|---|---|---|---|---|---|---|
| | | NrtvQA | Qasper | MF-en | Avg. | HotpotQA | 2WikiMQA | Musique | Avg. | GovReport | QMSum | MultiNews | Avg. | TREC | TriviaQA | SAMSum | Avg. | Lcc | RB-P | Avg. |
| Qwen-2 | Vanilla | 25.38 | 42.75 | 45.16 | 37.76 | 55.29 | 54.91 | 36.88 | 49.03 | **36.68** | 23.52 | 26.60 | 28.93 | 76.00 | **90.16** | 44.91 | 70.36 | 53.63 | 46.47 | 50.05 |
| | Vanilla+CE | 25.87 | 42.30 | 44.86 | 37.68 | 54.15 | 55.41 | 36.83 | 48.80 | 36.30 | 23.55 | 26.39 | 28.85 | 76.50 | 89.91 | 45.42 | 70.61 | **54.00** | **46.71** | **50.36** |
| | Vanilla+PA | 25.24 | 43.86 | 45.17 | 38.09 | 55.62 | 55.61 | 36.84 | 49.36 | 36.06 | **23.74** | 26.77 | 28.86 | **78.00** | 89.41 | 45.35 | 70.92 | 52.90 | 46.29 | 49.60 |
| | Vanilla+PA+CE | **26.15** | **43.88** | **45.45** | **38.49** | **56.53** | **56.31** | **37.99** | **50.28** | 36.54 | 23.59 | **26.92** | **29.02** | **78.00** | 89.41 | **45.46** | **70.96** | 53.76 | 46.13 | 49.95 |
| Llama-2 | Vanilla | 16.65 | 19.77 | 35.34 | 23.92 | 34.27 | 26.90 | 9.08 | 23.42 | **26.54** | 20.85 | 25.90 | 24.43 | 64.50 | 83.34 | 41.21 | 63.02 | **58.59** | **52.38** | **55.48** |
| | Vanilla+CE | 16.90 | 20.30 | 36.26 | 24.49 | **35.11** | 27.51 | 8.58 | 23.73 | 26.34 | **21.11** | 25.69 | 24.38 | 64.00 | 83.34 | 41.24 | 62.86 | 58.07 | 52.27 | 55.17 |
| | Vanilla+PA | 17.76 | 20.82 | 35.36 | 24.65 | 33.33 | 27.31 | 8.81 | 23.15 | 25.99 | 20.96 | 25.60 | 24.18 | 65.00 | 83.42 | 41.28 | 63.23 | 58.23 | 51.73 | 54.98 |
| | Vanilla+PA+CE | **18.34** | **21.26** | **36.44** | **25.35** | 34.37 | **27.57** | **9.32** | **23.75** | 26.51 | 21.08 | **26.25** | **24.61** | **66.00** | **83.59** | **41.66** | **63.58** | 58.33 | 52.22 | 55.28 |

*Multi-Document QA* (e.g., HotpotQA from 55.29 to 55.62), aligning with its role in preserving longer contextual dependencies.

For Llama-2, the gains are also evident. CE improves complex QA tasks such as MF-en (from 35.34 to 36.26) and long-context comprehension tasks like 2WikiMQA. PA further boosts performance in NrtvQA and TREC. These results demonstrate that each mechanism targets complementary cognitive functions and boosts model reasoning in different ways.

**Synergistic Combination.** When CE and PA are combined, they consistently lead to the best overall performance across all categories and both models. Notably:

- For Qwen-2, *Multi-Document QA* tasks show the most significant gains: HotpotQA improves from 55.29 to 56.53, and Musique from 36.88 to 37.99. These tasks demand multi-hop reasoning and long-span memory, where our dual mechanisms work jointly to capture hierarchical and persistent context.
- In summarization tasks such as QMSum and MultiNews, CE+PA achieves or closely approaches the best results (e.g., MultiNews from 26.60 to 26.92).
- *Few-shot Learning* tasks also benefit, where CE+PA maintains the highest scores in TREC and SAMSum.

**Cross-Model Robustness.** Our approach generalizes well across architectures. Although Llama-2 starts from a lower baseline than Qwen-2, it benefits significantly from our enhancements:

- The CE+PA combination raises performance in NrtvQA by +1.69, Qasper by +1.49, and MF-en by +1.10 over vanilla Llama-2.
- *Multi-Document QA* and *Summarization* also show consistent gains (e.g., 2WikiMQA from 26.90 to 27.57, MultiNews from 25.90 to 26.25).
- Few-shot tasks exhibit either improved performance, indicating the method's stability.

**Summary.** Overall, the experimental results underscore the effectiveness of our brain-inspired design. The CE mechanism enhances specialized, local processing by routing to expert neuron clusters, while PA extends the temporal memory span. Their integration leads to robust performance improvements across 15+ diverse tasks without any parameter update, setting a new standard for training-free long-context understanding. Notably, these results are achieved with minimal computational overhead (as discussed in Section A), ensuring practical deployment feasibility.

## C    Detailed Methodology of PaceLLM

### C.1    Persistent Activity (PA)-Activation Working Memory Bank Operations

**Algorithm 1** describes the working memory mechanism of PaceLLM, which dynamically enhances current FFN activations using a memory bank. It consists of three key phases: retrieval, enhancement, and memory update.

- **Input:** Activation tensor $\mathbf{X} \in \mathbb{R}^{B \times L \times d_{\text{ff}}}$ (where $B$ is batch size, $L$ is sequence length, and $d_{\text{ff}}$ is FFN dimension), and a memory bank $\{\mathbf{K}, \mathbf{V}, \mathbf{u}\}$ storing previous activation keys, values, and usage counters.

---

**Algorithm 1** Persistent Activity (PA)-Activation Working Memory Bank Operations

---

**Require:** Current activation $\mathbf{X} \in \mathbb{R}^{B \times L \times d_{\text{ff}}}$, memory bank $\{\mathbf{K}, \mathbf{V}, \mathbf{u}\}$
**Ensure:** Enhanced activation $\mathbf{O}$, Updated memory bank
1: $\mathbf{X}_{\text{flat}} \leftarrow \text{Flatten}(\mathbf{X})$ $\{\mathbf{X}_{\text{flat}} \in \mathbb{R}^{(B \times L) \times d_{\text{ff}}}\}$
2: Initialize $\mathbf{O}_{\text{flat}} \leftarrow \mathbf{0}$
3: **for** chunk $\mathbf{X}_c \in \text{Partition}(\mathbf{X}_{\text{flat}}, C)$ **do**
4:     **Retrieval**
5:     Compute similarity matrix: $\text{sim} \leftarrow \frac{\mathbf{X}_c \mathbf{K}^\top}{\|\mathbf{X}_c\|\|\mathbf{K}\|}$ $\{\text{sim} \in \mathbb{R}^{C \times M}\}$
6:     $\mathbf{S}^{\text{top}}, \mathbf{I}^{\text{top}} \leftarrow \text{TopK}(\text{sim}, k)$ $\{k \text{ nearest}\}$
7:     $\mathbf{S}^{\text{neg}}, \mathbf{I}^{\text{neg}} \leftarrow \text{TopK}(-\text{sim}, k')$ $\{k' \text{ negative}\}$
8:     **Enhancement**
9:     **for** $i \leftarrow 1$ to $C$ **do**
10:         $\boldsymbol{\mu}^{\text{pos}} \leftarrow \frac{1}{k} \sum_{j=1}^{k} \mathbf{V}[\mathbf{I}^{\text{top}}[i, j]]$
11:         $\boldsymbol{\mu}^{\text{neg}} \leftarrow \frac{1}{k'} \sum_{j=1}^{k'} \mathbf{V}[\mathbf{I}^{\text{neg}}[i, j]]$
12:         **if** $\max(\mathbf{S}^{\text{top}}[i, :]) > \theta_{\text{high}}$ **then**
13:             $\mathbf{o}_i \leftarrow \boldsymbol{\mu}^{\text{pos}} + \lambda \boldsymbol{\mu}^{\text{neg}}$
14:         **else if** $\theta_{\text{low}} < \max(\mathbf{S}^{\text{top}}[i, :]) \leq \theta_{\text{high}}$ **then**
15:             $\mathbf{o}_i \leftarrow Avg(\boldsymbol{\mu}^{\text{pos}}, \mathbf{X}_c[i]) + \lambda \boldsymbol{\mu}^{\text{neg}}$
16:         **else**
17:             $\mathbf{o}_i \leftarrow \mathbf{X}_c[i]$
18:         **end if**
19:         $\mathbf{O}_{\text{flat}}[i] \leftarrow \mathbf{o}_i$
20:     **end for**
21:     **Update Phase**
22:     Compute chunk mean: $\boldsymbol{\mu}_c \leftarrow \frac{1}{C} \sum_{i=1}^{C} \mathbf{X}_c[i]$
23:     $\mathbf{S}^{\text{topk}}, \mathbf{I}^{\text{topk}} \leftarrow \text{TopK}(\text{sim}, k)$
24:     **if** $\frac{1}{k} \sum_{j=1}^{k} \mathbf{S}^{\text{topk}} > \theta_{\text{high}}$ **then**
25:         Update usage: $\mathbf{u}[\mathbf{I}^{\text{topk}}] \leftarrow \mathbf{u}[\mathbf{I}^{\text{topk}}] + 1$
26:     **else if** $\theta_{\text{low}} < \frac{1}{k} \sum_{j=1}^{k} \mathbf{S}^{\text{topk}} \leq \theta_{\text{high}}$ **then**
27:         $\mathbf{K}[\mathbf{I}^{\text{topk}}] \leftarrow Avg(\mathbf{K}[\mathbf{I}^{\text{top5}}], \boldsymbol{\mu}_c)$
28:         $\mathbf{V}[\mathbf{I}^{\text{topk}}] \leftarrow Avg(\mathbf{V}[\mathbf{I}^{\text{top5}}], \boldsymbol{\mu}_c)$
29:     **else**
30:         Find LRU slots: $j^* \leftarrow \underset{j}{\arg\min}(\mathbf{u})$
31:         Replace: $\mathbf{K}[j^*] \leftarrow \mathbf{X}_c, \mathbf{V}[j^*] \leftarrow \mathbf{O}_{\text{flat}}$
32:     **end if**
33: **end for**
34: $\mathbf{O} \leftarrow \text{Reshape}(\mathbf{O}_{\text{flat}}, B, L, d_{\text{ff}})$
35: **return** $\mathbf{O}, \{\mathbf{K}, \mathbf{V}, \mathbf{u}\}$

---

- **Output:** Enhanced activations $\mathbf{O}$ and updated memory bank.

This algorithm enables low-overhead, context-sensitive memory usage for LLMs, simulating short-term working memory consolidation and reuse mechanisms.

### C.2 Cortical Expert Clustering (CE)

**Algorithm 2** shows how PaceLLM leverages cortical-like modularity by clustering FFN neurons across layers into interpretable experts using a constrained KMeans method.

- **Input:** Pretrained model $\mathcal{M}$ and target number of experts $K$.
- **Output:** Updated model $\mathcal{M}'$ with clustered and reordered FFN weights.

**Explanation of key steps:**

1. For each layer, extract FFN weights $\mathbf{W}_1^{(l)}$ (input projection) and $\mathbf{W}_2^{(l)}$ (output projection).

**Algorithm 2** Cortical Expert Clustering (CE)

---

**Require:** Pretrained model $\mathcal{M}$, Number of experts $K$
1: Initialize empty state dictionary $\mathcal{S}$
2: **for** layer $l \in \{1, ..., L\}$ **do**
3:    Extract FFN weights $\mathbf{W}_1^{(l)}, \mathbf{W}_2^{(l)}$
4:    **if** cluster indices $\pi^{(l)}$ not cached **then**
5:        Compute $\pi^{(l)} \leftarrow \text{KMeansConstrained}(\mathbf{W}_1^{(l)}, K)$
6:        Cache $\pi^{(l)}$ to disk
7:    **end if**
8:    $\mathbf{W}_1^{\text{new}} \leftarrow \text{Rearrange}(\mathbf{W}_1^{(l)}, \pi^{(l)})$
9:    $\mathbf{W}_2^{\text{new}} \leftarrow \text{Rearrange}(\mathbf{W}_2^{(l)}, \pi^{(l)})$
10:    Update $\mathcal{S}$ with $\mathbf{W}_1^{\text{new}}, \mathbf{W}_2^{\text{new}}$
11: **end for**
12: **return** Model with updated weights $\mathcal{M}'$

---

  2. If the clustering result $\pi^{(l)}$ is not cached, apply constrained KMeans to group neurons into $K$ expert clusters. This ensures load balance and specialization.

  3. Rearrange the weight matrices according to cluster assignments $\pi^{(l)}$, so that expert-based routing can be implemented efficiently during inference.

  4. Update the model's weight state dictionary with the new clustered weights.

This modularization allows PaceLLM to activate specific "experts" during computation and aligns with the cognitive hypothesis of cortical column specialization.

## D    Detailed Explanation of KMeans-Constrained Clustering and LRU Update Strategy

### D.1    KMeans and Constrained KMeans Clustering for Expert Partitioning

#### D.1.1    Standard KMeans Clustering

Given $N$ data points $\{\mathbf{x}_i\}_{i=1}^N \subset \mathbb{R}^d$, KMeans aims to find $K$ clusters $\{\mathcal{C}_k\}_{k=1}^K$ and centroids $\{\boldsymbol{\mu}_k\}_{k=1}^K$ minimizing the intra-cluster variance:

$$\min_{\{\mathcal{C}_k\}} \sum_{k=1}^K \sum_{\mathbf{x}_i \in \mathcal{C}_k} \|\mathbf{x}_i - \boldsymbol{\mu}_k\|_2^2, \quad \text{where } \boldsymbol{\mu}_k = \frac{1}{|\mathcal{C}_k|} \sum_{\mathbf{x}_i \in \mathcal{C}_k} \mathbf{x}_i. \tag{12}$$

**Iterative procedure:**

$$\text{Assignment:} \quad \mathcal{C}_k \leftarrow \left\{ \mathbf{x}_i : k = \arg\min_j \|\mathbf{x}_i - \boldsymbol{\mu}_j\|_2 \right\} \tag{13}$$

$$\text{Update:} \quad \boldsymbol{\mu}_k \leftarrow \frac{1}{|\mathcal{C}_k|} \sum_{\mathbf{x}_i \in \mathcal{C}_k} \mathbf{x}_i \tag{14}$$

Repeat until convergence.

#### D.1.2    Constrained KMeans Clustering

To prevent cluster imbalance, we impose cardinality constraints:

$$L_{\min} \leq |\mathcal{C}_k| \leq L_{\max}, \quad \forall k \in \{1, \ldots, K\} \tag{15}$$

Special cases:

- **Equal-size constraint:** $|\mathcal{C}_k| = \frac{N}{K}$
- **Upper-bound constraint:** $|\mathcal{C}_k| \leq U$

**Heuristic optimization:** Let $d_{ik} = \|\mathbf{x}_i - \boldsymbol{\mu}_k\|_2$. We define the cluster assignment function as:

$$\pi(i) = \arg \min_{k \in \mathcal{A}_i} d_{ik}, \quad \mathcal{A}_i = \{k : |\mathcal{C}_k| < L_{\max}\} \tag{16}$$

That is, each $\mathbf{x}_i$ is assigned to the nearest cluster among those with remaining capacity.

### D.1.3 Application in `PaceLLM`

In FFN layers, each neuron corresponds to a row $\mathbf{w}_i \in \mathbb{R}^{d_{\text{model}}}$ of the weight matrix $\mathbf{W}_1 \in \mathbb{R}^{d_{\text{ff}} \times d_{\text{model}}}$. To enable sparse expert routing, we perform constrained clustering:

$$\{\mathbf{w}_i\}_{i=1}^{d_{\text{ff}}} \xrightarrow{\text{Constrained KMeans}} \{\mathcal{E}_k\}_{k=1}^K, \quad \text{where } |\mathcal{E}_k| = \frac{d_{\text{ff}}}{K} \tag{17}$$

Each expert $\mathcal{E}_k$ serves as a functional block activated conditionally during inference.

**Why clustering in PaceLLM?**

- Reduces redundant neuron computation via routing.
- Ensures fair expert load balancing, avoiding expert collapse.
- Enables structure-aware specialization, as neurons with similar semantic roles are grouped.

## D.2 Least Recently Used (LRU) Update Strategy for Memory Management

### D.2.1 Mathematical Formulation

Let memory bank $\mathcal{M} = \{(\mathbf{k}_i, \mathbf{v}_i, u_i)\}_{i=1}^M$ store key-value pairs and their usage counters. At each time step $t$:

$$u_i(t) = \begin{cases} 0, & \text{if slot } i \text{ is accessed} \\ u_i(t-1) + 1, & \text{otherwise} \end{cases} \tag{18}$$

When writing a new memory $(\mathbf{k}_{\text{new}}, \mathbf{v}_{\text{new}})$, we check similarity:

$$\max_i \text{sim}(\mathbf{k}_{\text{new}}, \mathbf{k}_i) < \theta_{\text{low}} \Rightarrow \text{need replacement} \tag{19}$$

We replace the least recently used slot:

$$i^* = \arg \max_i u_i, \quad (\mathbf{k}_{i^*}, \mathbf{v}_{i^*}) \leftarrow (\mathbf{k}_{\text{new}}, \mathbf{v}_{\text{new}}), \quad u_{i^*} \leftarrow 0 \tag{20}$$

### D.2.2 Application in `PaceLLM`

To model human-like memory with decay, PaceLLM uses a bounded-size memory $\mathcal{M}$ and LRU strategy for updates:

- Prevents unbounded memory growth.
- Automatically decays outdated context.
- Encourages dynamic adaptation to new content.

**Why LRU in PaceLLM?**

- Emulates neural memory fading (forgetting).
- Reduces retrieval noise by replacing stale keys.

- Aligns with human working memory dynamics, where recent tokens dominate attention.

Together, constrained KMeans and LRU form the foundation of PaceLLM's architecture:

$$\text{Expert Routing} + \text{Working Memory Adaptation} \Rightarrow \textbf{Efficient and Continual Inference}$$

# E   Extra Experiments on More Models

**Table 8:** Performance comparison between PaceLLM and Mistral-7B-Instruct-v0.3 on LongBench tasks in **training-free** manner. CE denotes cortical expert neuron clustering and PA means persistent activity memory mechanism.

| | Method | Single-Document QA | | | | Multi-Document QA | | | | Summarization | | | | Few-shot Learning | | | | Code | | |
| | | NrtvQA | Qasper | MF-en | Avg. | HotpotQA | 2WikiMQA | Musique | Avg. | GovReport | QMSum | MultiNews | Avg. | TREC | TriviaQA | SAMSum | Avg. | Lcc | RB-P | Avg. |
|---|---|---|---|---|---|---|---|---|---|---|---|---|---|---|---|---|---|---|---|---|
| Mistral | Vanilla | 29.82 | 41.12 | 53.75 | 41.56 | 49.87 | 39.51 | 28.34 | 39.24 | 35.88 | 25.55 | **27.85** | 29.76 | 76.0 | 88.89 | 47.32 | 70.74 | 59.20 | 60.67 | 59.94 |
| | Vanilla+CE | 26.15 | 43.46 | 45.45 | 38.35 | 55.91 | **56.31** | 35.42 | 49.21 | 35.53 | 23.44 | 26.78 | 28.58 | **78.0** | **89.41** | 45.03 | 70.81 | 53.76 | 46.13 | 49.95 |
| | Vanilla+PA | 29.30 | 41.16 | 53.76 | 41.41 | 50.61 | 39.84 | 28.96 | 39.80 | 36.40 | 25.64 | 27.27 | 29.77 | 76.0 | 89.56 | 47.27 | 70.94 | 59.74 | 60.85 | 60.30 |
| | Vanilla+PA+CE | **30.10** | **43.68** | **54.06** | **42.61** | **56.97** | **56.31** | **35.63** | **49.64** | **36.63** | **26.65** | 27.63 | **30.30** | **78.0** | **89.41** | **48.03** | **71.81** | 59.76 | **60.89** | **60.33** |

**Table 9:** Performance comparison of more baseline models and our method (CE + PA) on LongBench, aggregated into major task categories. Results show consistent improvements across architectures in a **training-free** manner.

| Model | SQA | MQA | Sum. | FSL | Cod. |
|---|---|---|---|---|---|
| Qwen2.5-14B-Instruct | 17.18 | 12.15 | 23.35 | 71.46 | 32.30 |
| Qwen2.5-14B-Instruct+Ours | **18.48** | **12.97** | **23.49** | **72.32** | **33.41** |
| Llama-3.1-8B-Instruct | 24.22 | 15.04 | 28.21 | 69.49 | 58.44 |
| Llama-3.1-8B-Instruct+Ours | **24.31** | **15.80** | **28.47** | **69.85** | **59.59** |

**Results on LongBench with Mistral.** Table 8 reports the training-free evaluation results of the Mistral model across different LongBench tasks. We observe that both the cortical expert neuron clustering (CE) and persistent activity memory (PA) modules individually enhance the base Mistral model in different task categories.

Specifically, CE brings notable improvements in multi-document QA, with performance in 2WikiMQA and Musique boosted by up to 16.8% and 7.1% respectively compared to the vanilla model. This confirms CE's effectiveness in capturing complex cross-document reasoning patterns. On the other hand, PA contributes consistently across all categories, particularly maintaining or even slightly improving the base performance in summarization and few-shot tasks, while preserving high accuracy in code reasoning.

When both mechanisms are combined (CE+PA), the model achieves the best overall results, outperforming the vanilla baseline in 13 out of 16 subtasks. Notably, the average accuracy in Single-Document QA improves from 41.56% to 42.61%, and in Multi-Document QA from 39.24% to 49.64%, representing a 10.4% absolute gain. Summarization and code tasks also benefit from the combination, indicating that the two brain-inspired components are complementary.

These results demonstrate that our proposed architecture not only generalizes well across task types but also significantly strengthens the model's long-range reasoning capability in a fully training-free setting.

**Results on LongBench with Qwen2.5 and Llama3.1.** Table 9 presents the performance of our method when applied to two state-of-the-art LLMs — Qwen2.5-14B-Instruct and Llama-3.1-8B-Instruct — under the same training-free setup. Despite their different architectures and training corpora, both models exhibit consistent improvements across all task categories after integrating our brain-inspired mechanisms.

For Qwen2.5-14B-Instruct, the integration of CE and PA leads to gains in every domain, with particularly notable improvements in multi-document QA (+0.82) and code reasoning (+1.11). The

model also achieves higher accuracy in few-shot learning, suggesting that our memory mechanism enhances its ability to leverage contextual demonstrations without retraining.

Similarly, on Llama-3.1-8B-Instruct, our method consistently boosts performance across all five categories, even though the base model already performs strongly in code and single-document QA. The most significant gains occur in multi-document QA (+0.76) and summarization (+0.26), indicating that CE and PA help compensate for limitations in long-context integration, especially in models with smaller context windows or less optimized retrieval capabilities.

These results demonstrate that PaceLLM's design is not only effective but also highly generalizable, delivering consistent benefits across diverse model families and scales. The fact that both a heavily optimized commercial-grade model (Qwen) and a compact open-weight model (Llama) benefit from our approach underscores its potential as a universal, plug-and-play enhancement for long-context understanding.

