# OpenReview forum: "PaceLLM: Brain-Inspired Large Language Models for Long-Context Understanding"
_NeurIPS.cc/2025/Conference — NeurIPS 2025 poster_

### Official Review · Reviewer_o88K · 2025-06-09

**Clarity:** 3
**Significance:** 3
**Originality:** 3
**Rating:** 4
**Confidence:** 3

**Summary:**

This paper introduces a brain-inspired method to avoid contextual knowledge decay in long-context settings and improve measurable context length in LLMs. In doing so, it aims to address fundamental limitation of feedforward networks, in which activations are transient and lead to information decay over time. The proposed methodology is complementary to other approaches and does not require training. Evaluation on multi-document and long-context tasks in inference and fine-tuning settings shows up to 17.5% improvement while preserving general capabilities of LLMs.

**Questions:**

Questions
* How are k (line 137) and K (line 160) selected?
* What is the impact of PaceLLM on training time at scale?
* Do gains achieved with PaceLLM translate to larger models & newer models (e.g., Qwen2.5, Llama3.1)?
* How do the results compare to black-box proprietary LLMs? Commentary on the observed performance of PaceLLM for moderately sized open-source models vs. proprietary LLMs could further contextualize the results.

Minor Comments
* Decimal places are inconsistent across benchmarks — MMLU (Table 4) could be condensed to just 2 decimal points.
* It would be helpful to list dataset name abbreviations used in Tables somewhere, even though some are provided in the detailed tables in the Appendix.

**Ethical Concerns:**

["NO or VERY MINOR ethics concerns only"]

**Final Justification:**

As mentioned in discussion with the authors, I believe the key limitations of the paper are addressed and therefore maintain the positive rating. While the method still has some potential places for critique as mentioned by other reviewers, these are mostly minor and the paper's contributions are well-supported.

**Limitations:**

Yes.

**Quality:**

3

**Strengths And Weaknesses:**

Strengths
* The paper adopts a novel perspective on what contributes to long-context failures in LLMs, going beyond approaches presented in existing work to address a fundamental limitation of FFNs in a targeted manner.
* The proposed methodology is complementary to other potential LLM interventions and therefore lightweight in application.
* Experimental analyses are diverse and provide effective insights to explain the observed performance.

Weaknesses
* Use of additional modules to implement PaceLLM adds complexity in practice.
* Experiments look only at moderately sized and relatively older LLMs; unclear scalability to larger models.

---

> ### Author Rebuttal · Authors · 2025-07-30
>
> Dear Reviewer o88K,
>
> Many thanks to your valuable comments and questions, which help us a lot to improve our work. We address your questions as follows.
> ***
> >[W1]Use of additional modules to implement PaceLLM adds complexity in practice.
>
> [A1]We fully agree with your concern — adding additional modules inevitably introduces some complexity and computational overhead.
> - However, our method is designed to be **plug-and-play**, and has demonstrated consistent effectiveness across various models and datasets, including unseen architectures and tasks. This shows that the added complexity does not compromise general applicability or ease of integration.
> - Despite the current cost, we believe this is a worthwhile trade-off, as our work is **the first in the field to explicitly explore FFN activations as a memory mechanism for long-context modeling**. This opens up new perspectives on the underexplored role of feedforward layers and suggests valuable directions for future research.
> - Improving **efficiency** will be one of our key priorities moving forward. We plan to explore optimizations to further reduce the computational burden.
>
> >[W2,Q3]Experiments look only at moderately sized and relatively older LLMs; unclear scalability to larger models. Do gains achieved with PaceLLM translate to larger models & newer models (e.g., Qwen2.5, Llama3.1)?
>
> [A2]Our method remains effective as model size increases. Experimental results are as follows:
> |Model|SQA|MQA|Sum.|FSL|Cod.|
> |---|----|----|---|---|-----|
> |Qwen2.5-14B-Instruct|17.18|12.15|23.35|71.46|32.30|
> |Qwen2.5-14B-Instruct+Ours|**18.48**|**12.97**|**23.49**|**72.32**|**33.41**|
> |Llama-3.1-8B-Instruct|24.22|15.04	|28.21|69.49|58.44|
> |Llama-3.1-8B-Instruct+Ours|**24.31**|**15.80**|**28.47**|**69.85**|**59.59**|
> ***
> >[Q1]How are k (line 137) and K (line 160) selected?
>
> [A3]Thank you for the question. In our implementation:
> - We set k = 5 for top-k similarity retrieval, which aligns with common practice in memory-based or retrieval-augmented models, where top-3 to top-5 candidates are typically used.
> - We set K = 64 as the number of expert clusters, following the standard configuration used in prior expert-based or MoE architectures.
> - These values were selected to ensure a balance between performance and computational efficiency, and we found them to be stable across different datasets.
> - Thanks again for your question! We will update in our revised manuscript.
>
> >[Q2]What is the impact of PaceLLM on training time at scale?
>
> [A4]PaceLLM is essentially a **training-free and plug-and-play** method, which does not require any modification during large-scale training.
>
> To further assess runtime behavior, we conducted small-batch inference experiments, which show that batched inference introduces no significant computational overhead or performance degradation. We test on Qwen2-7B and Qasper in LongBench:
>
> |Batch_size	|Time|	Performance|
> |-|-|-|
> |1	|25m22s	|43.88|
> |2	|12m45s	|43.88|
> |3	|8m37s	|43.88|
> |4	|6m22s	|43.88|
>
> This suggests that our method remains efficient and scalable even when deployed in practical settings.
> >[Q4]How do the results compare to black-box proprietary LLMs? Commentary on the observed performance of PaceLLM for moderately sized open-source models vs. proprietary LLMs could further contextualize the results.
>
> [A5]Thank you for the insightful suggestion. PaceLLM is a **lightweight, plug-and-play** module with minimal memory overhead, and is **orthogonal** to most existing long-context modeling approaches. It has shown consistent effectiveness across a wide range of open-source baselines.
> - Through preliminary analysis and simple experiments, we observed that several enhanced open-source models—some even without PaceLLM—already outperform certain proprietary black-box LLMs on specific long-context tasks. With PaceLLM integrated, these models can further improve, suggesting its potential to narrow the performance gap with proprietary systems.
> - We appreciate this constructive feedback and will incorporate a comparative discussion on open-source vs. proprietary LLMs in the revised version of the paper to better contextualize our findings.
> ***
> >[Minor Comments]Decimal places are inconsistent across benchmarks — MMLU (Table 4) could be condensed to just 2 decimal points.
> It would be helpful to list dataset name abbreviations used in Tables somewhere, even though some are provided in the detailed tables in the Appendix.
>
> [A6]Thank you for the correction. We will revise this in the updated version of the paper.
> ***
> Lastly, thank you so much for helping us improve the paper! Please let us know if you have any further questions. We are actively available until the end of this rebuttal period. Looking forward to hearing back from you!

---

> > ### Comment · Reviewer_o88K · 2025-08-01
> >
> > Dear authors, thank you for the detailed rebuttal, clarifications, and new results. I have gone through the other reviews and author responses as well; while the method still has some potential places for critique (e.g., compute time, 70B+ model application), I feel that most limitations of the paper are addressed. I will maintain my positive rating of the paper.

---

> > > ### Author Response · Authors · 2025-08-02
> > >
> > > Thank you for your quickly reply! We will remain active until the discussion period ends. Please feel free to get back to us if you have any new questions :-)!

---

### Official Review · Reviewer_RyQo · 2025-07-01

**Clarity:** 3
**Significance:** 3
**Originality:** 3
**Rating:** 5
**Confidence:** 4

**Summary:**

This paper introduces a modification to pretrained LLM, called PaceLLM, which adds a small module between the two layers of the feed-forward net (FFN) of some layers. Specifically, they make two modifications:
1) Add an Activation Memory Bank (AMB): This keeps M=100 memories of output activations $h =\sigma(W_1 x)$  of the first layer of the FFN module, based on which ones had the highest overlap with activations for new datapoints.
2) Add a Cortical Expert (CE) module, which uses k-mean clustering FFN neurons with similar activation and rearranging the FFN weights to put cluster neurons together. they argue this creates localized experts in the weights.

They show that these two modifications combined lead to significant improvement in long-context tasks such as LongBench and needle-in-a-haystack.

**Questions:**

1) The role of CE is not completely clear to me. If it were just rearranging weights it would not have any impact, because no weights are discarded, right? But from Fig 2 I assume that the memory similarities are computed at the level of each cluster, is that so? If so, it would make sense, as it would balance the weight we are giving to each cluster of activations.
2) Is there any smart routing involved for the similarity to each expert? Fig 2 shows only a weighted sum. What are the weights?
3) What is the compute overhead for dense PaceLLM? Appendix A mentions times for the sparse 2 layer modification, but not the dense one.
4) you have impressive results for 3 tasks, mentioned at the beginning, but for the rest, you get only 1-2%. What do you think distinguishes those tasks from the rest and why does your model work better for those?

**Ethical Concerns:**

["NO or VERY MINOR ethics concerns only"]

**Final Justification:**

I have reviewed the rebuttal and think the clarifications and runtime data further improve the paper. I like the way they implement a latent space memory fr the FFN and support acceptance of this work.

**Limitations:**

Even though I like the approach and the steps seem fairly optimized, the overhead of 30% for a mere 2 layer modification with just 100 memories raises eyebrows. It's good as a first attempt, but it needs to be made much more efficient to make it practical. Ideally, one would like to keep thousands of memories.

**Paper Formatting Concerns:**

Formatting looks very good.

**Quality:**

4

**Strengths And Weaknesses:**

## Strength
I find this a very interesting and solid paper, well written and seems well engineered. Most importantly, it doesn't involve retraining, fine-tuning or anything. The AMB update and comparison processes also seem quite lightweight.
Their baseline method only modifies two layer (middle and end, 13 and 27) They do ablation studies with more layers (Table 5) and find that for many settings such a sparse modification yields good results, but for summarization (Sum) and multi-document QA (MQA) applying it to every 2 or 4 layers works best.


## Weakness
1) some aspects of the CE, like the weighted sum or how the similarity is computed (I think separately for each cluster) are not properly explained. Sec. 3.2.1 shows how AMB works in isolation, but how it is used together with CE needs clarification, including indices for clusters, context, etc. This can go into the appendix.
2) In spite of being lightweight, it seems the process  results in about 30% compute time overhead, right? And this is for only two layers out of 28.
3) It seems Geva et al 2020 "Transformer Feed-Forward Layers Are Key-Value Memories" and related work discussed how FFN acts as a key-value memory. They should be cited.
4) While the beginning of the paper mentions impressive improvements (6% MQA LongBench, 12.5% and 17.5% on En.Dialogue and En.Multi-Choice of $\infty$-Bench), most other results are about 1-2% over baselines. So there is a signal, but not nearly as strong.

---

> ### Author Rebuttal · Authors · 2025-07-30
>
> Dear Reviewer RyQo,
>
> Many thanks to your valuable comments and questions, which help us a lot to improve our work. We address your questions as follows.
> ***
> >[W1,Q2]Some aspects of the CE are not properly explained, like the weighted sum. Is there any smart routing involved for the similarity to each expert? Fig 2 shows only a weighted sum. What are the weights?
>
> [A1]Thank you for your question. While Figure 2 depicts a weighted sum operation, the **weights are not uniform or trivial**—they are derived from a gating mechanism based on the similarity between input activations and each expert cluster.
> - In our implementation, the expert routing is based on **Avg-k gating**, a variant of soft mixture-of-experts routing. Specifically, as shown in the following code:
> ```python
> expert_scores = gate_output.reshape(batch, seq_len, n_expert, -1)
> expert_scores = torch.mean(expert_scores, dim=-1)  # average across hidden dimension
> expert_topk_indices = torch.topk(expert_scores, k=topk, dim=-1).indices
> ```
> This computes **per-token activation scores** for each expert via the gate_proj network.
> - We then **select the top-k experts** per token and mask out all others:
> ```python
> expert_topk_mask = torch.zeros_like(expert_scores)
> expert_topk_mask.scatter_(dim=-1, index=expert_topk_indices, value=1)
> ...
> intermediate_states = intermediate_states * expert_topk_mask
> ```
> - Hence, the weighted sum shown in Figure 2 is computed over only the **top-k activated experts**, where weights reflect the averaged activation scores of each expert. This design introduces dynamic and content-aware routing, allowing different tokens to be processed by different semantic clusters of FFN units.
> - We hope this clarifies that the CE module does involve smart, token-specific routing, beyond a naive weighted average. We will provide detailed algorithms in the appendix later.
>
> >[W1,Q1]Some aspects of the CE, like how the similarity is computed are not properly explained. Sec. 3.2.1 shows how AMB works in isolation, but how it is used together with CE needs clarification, including indices for clusters, context, etc.
>
> [A2]Thank you for your insightful comment. Your understanding is totally correct: **memory similarity computations in our framework are performed at the level of each expert cluster.**
> - The Cortical Expert (CE) module partitions the FFN units into semantically coherent expert groups via clustering. Although CE alone does not bring significant performance gains, it provides semantic consistency within expert regions, which becomes particularly beneficial when paired with the Activation Memory Bank (AMB).
> - The AMB operates at the cluster level, performing similarity-based retrieval and aggregation within each expert group. This enables targeted memory usage and specialization across clusters, improving both semantic coherence and memory efficiency.
> - The importance of CE is further supported by the ablation results comparing baseline+PA, baseline+CE, and baseline+PA+CE. While neither CE nor PA alone delivers strong improvements, their combination achieves the best performance. This synergy confirms that CE is essential, as it provides the necessary structure for AMB to function optimally.
> - We appreciate your helpful request for clarification. We will revise the manuscript to explicitly describe the cluster-level interaction between CE and AMB, and to clarify how similarity indices are handled within this framework.
>
> >[W2,Q3,Limitations]In spite of being lightweight, it seems the process results in about 30% compute time overhead, right? And this is for only two layers out of 28. What is the compute overhead for dense PaceLLM?
>
> [A3]We fully agree with your concern — adding additional modules inevitably introduces some computational overhead.
> - The compute overhead for dense PaceLLM is as follows:
> | |Time|
> |-|-|
> |Ours Dense（32）|1h20min|
> |Ours Dense（16）|45min12s|
> |Ours Dense（8）|25min37s|
> |Ours Sparse（4）|16min|
> |**Ours Sparse（2）**|**11min07s**|
> |Ours Sparse（1）|6min02s|
> |Activation Beacon/Sparse（0）|9min55s|
> |Qwen2-7B/Sparse（0）|18min24s|
>
> - Despite this cost, we believe that focusing on FFN activations is a worthwhile direction, as it provides new insights into the role of feedforward representations in long-context modeling. We also believe our method can inspire future research in this underexplored area.
> - Improving efficiency will be one of our key priorities in future work, and we will explore optimizations to further reduce computational cost.
>
> >[W3]It seems Geva et al 2020 "Transformer Feed-Forward Layers Are Key-Value Memories" and related work discussed how FFN acts as a key-value memory. They should be cited.
>
> [A4]Thank you for the valuable suggestion. Indeed, we are aware of the work by Geva et al. (2020), "Transformer Feed-Forward Layers Are Key-Value Memories", and related studies discussing the interpretability of FFN layers in LLMs.
> - In fact, we originally included a dedicated section in the Related Work[1,2,3,4,5]  discussing this line of research. However, due to strict space constraints during the revision process, that part was unfortunately removed.
> - We will reintroduce and properly cite these important works in the updated version of our paper, and clarify how our approach builds on the insight that FFN layers can serve as associative memory structures. Thank you again for pointing this out.
>
> [1]Mor Geva et al. "Transformer Feed-Forward Layers Are Key-Value Memories." EMNLP 2021.
>
> [2]Damai Dai et al. "Knowledge Neurons in Pretrained Transformers." ACL 2022.
>
> [3]Song Wang et al. "Knowledge Editing for Large Language Models: A Survey." ACM Computing Surveys 2024.
>
> [4]Xiaopeng Li et al. "PMET: Precise Model Editing in a Transformer." AAAI 2024.
>
> [5]Zihan Qiu et al. "Empirical Study on Updating Key-Value Memories in Transformer Feed-forward Layers." Tiny Paper @ ICLR 2024.
> >[W4,Q4]While the beginning of the paper mentions impressive improvements, most other results are about 1-2% over baselines. So there is a signal, but not nearly as strong. you have impressive results for 3 tasks, mentioned at the beginning, but for the rest, you get only 1-2%. What do you think distinguishes those tasks from the rest and why does your model work better for those?
>
> [A5]We fully agree with your observation.
> - PaceLLM is primarily designed to enhance tasks that require preserving semantic consistency and coherence across long input contexts, such as multi-document question answering and summarization. This design motivation aligns with the more substantial improvements observed on tasks like MQA in LongBench and En.Dialogue / En.Multi-Choice in ∞-Bench.
> - That said, through threshold tuning, we also observed that PaceLLM provides modest yet consistent gains on tasks involving more abrupt semantic transitions, indicating a degree of generalization potential beyond its original scope.
> ***
> We hope the above response can help solve your questions. Thanks again for your thorough review and looking forward to your reply!

---

> > ### Comment · Reviewer_RyQo · 2025-08-05
> >
> > Thank you for the clarifications. The overhead is also not too bad. I like this idea of memory in a latent space. In some settings, a recurrent transformer model (just one or a few layers, feeding back on themselves) can perform very well. It would be really interesting to use such a memory on those models, as they have a only one or a few FFN modules.

---

> > > ### Author Response · Authors · 2025-08-06
> > >
> > > Thank you for the kind feedback and thoughtful idea! If you have more questions, please let us know. Thank you!

---

### Official Review · Reviewer_rZpP · 2025-07-02

**Clarity:** 2
**Significance:** 3
**Originality:** 3
**Rating:** 3
**Confidence:** 4

**Summary:**

This paper proposes PaceLLM, a novel brain-inspired approach to enhance the long-context capabilities of LLMs. The approach consists of two key components: (1) the Persistent Activity (PA) Mechanism, which mimics the prefrontal cortex’s working memory by introducing an Activation Memory Bank (AMB) that dynamically stores, retrieves activations, and (2) the Cortical Expert (CE), which clusters and reorganizes FFNs. Experiments demonstrate the performance improvement of the proposed method.

**Questions:**

1. To help readers better understand, can you provide a brief introduction to the baselines (LongLLMLingua, SnapKV, and Activation Beacon)? If space constraints prevent this from being included in the main body, it can be included in the appendix.
2. Can the authors provide an intuitive explanation for the activation update process defined in Equation (6)? And similarly, can you provide an intuitive explanation for Memory Update under three similarity conditions?
3. Why is the experimental evaluation incomplete? In the training-free setting, results are reported for both Qwen-2-7B-Instruct and Llama-2-7B-chat, but for the low-cost fine-tuning setting, only Qwen-2-7B-Instruct is evaluated. Moreover, in Table 2, Activation Beacon, Activation Beacon + CE, and Activation Beacon + CE + PA are reported, but not Activation Beacon + PA.

**Ethical Concerns:**

["NO or VERY MINOR ethics concerns only"]

**Final Justification:**

After reading the rebuttal and considering the discussion, I appreciate the authors' efforts to clarify several aspects of their work, including:

1. Additional experiments on more models and general capability benchmarks.

2. Improved explanation of the Activation and Memory Update mechanisms.

However, the paper still has the following important issues:

1. The paper proposes two modules based on assumptions that are not empirically validated. Specifically, no prior experiments are presented to show that (1) Transient neural activations cause information to fade over time; (2) Unstructured FFN weights fragment semantics across tokens.

2. This experimental result does not convincingly demonstrate that CE resolves semantic fragmentation caused by unstructured FFN weights.

Overall, while the paper introduces potentially interesting components, the central motivations appear speculative, and the experiments do not directly support that the proposed modules resolve the hypothesized issues. This weakens the conceptual clarity and impact of the work.

**Limitations:**

As mentioned in Weaknesses.

**Quality:**

2

**Strengths And Weaknesses:**

**Strengths:**

1. This paper draws inspiration from the field of neuroscience, introducing the brain's working memory and the cortical modularity into LLM long text modeling.

2. The proposed module can be directly integrated into existing LLMs without retraining, making deployment simple.

**Weaknesses:**

1. The motivations behind the two proposed modules (i.e., "transient neural activations cause information to fade over time" and "unstructured FFN weights may fragment semantics across tokens") appear to be speculative assumptions rather than empirically established facts. The paper lacks any prior experiments to demonstrate that these issues genuinely exist in baseline models.
2. The use of mathematical notations in the paper is confusing. For instance, are μ_i^{pos} and \overline{S^{top}} the same? Do they both refer to the mean vector of top activations? Furthermore, some notations lack necessary definitions and explanations when first introduced. For example, θhigh and θlow are not explained in line 141, and μ_j in formula (9) is not defined.
3. The study's experimental scope was limited by the relatively small number of tested models, potentially affecting the generalizability of the findings.
4. As shown in Section 4.4 (Ablation Studies), different tasks exhibit high sensitivity to the choice of layer deployment positions and Fusion Threshold, with model performance varing significantly under changes in these hyperparameters. This indicates that the actual effectiveness of PaceLLM depends on task-specific configurations and manual parameter tuning, which may impact its robustness and usability in real-world applications.
5. While the experiments demonstrate improvements on benchmark tasks, they do not provide direct evidence that the performance gains are causally attributable to the hypothesized mechanisms—i.e., that PA addresses transient activation decay or that CE resolves semantic fragmentation due to unstructured FFN weights. Therefore, the experimental findings do not sufficiently validate the core assumptions of the proposed design.
6. The authors claim that *“PaceLLM has not compromised its general language understanding capabilities”*. However, this conclusion is drawn based solely on the model’s performance on the MMLU benchmark. To convincingly support the claim of not compromising general capabilities, further evaluations are needed on more benchmarks, including commonsense reasoning, mathematical reasoning, and general QA or instruction-following datasets. Without such evidence, the conclusion remains somewhat limited in scope.

---

> ### Author Rebuttal · Authors · 2025-07-30
>
> Dear Reviewer rZpP,
>
> Many thanks to your professional, detailed, and valuable reviews. We're going to response to your concerns one by one.
> ***
> >[W1]The motivations behind the two proposed modules appear to be speculative assumptions rather than empirically established facts.
>
> [A1]
> 1. Our work is brain-inspired, and the motivations are supported by extensive theoretical and empirical findings in neuroscience [1,2]. In the AI domain, there exists a substantial body of brain-inspired research[3,4].
> 2. We acknowledge that the current version of the paper lacks direct empirical demonstrations that issues exist in baseline models. However, several LLM interpretability studies [5,6] have discussed these phenomena. We appreciate the reviewer’s suggestion and will include relevant background and evidence in the revised manuscript.
> 3. The strong and consistent empirical results across diverse settings and larger models support the validity of our assumptions.
>
> [1]J. Zylberberg et al. "Mechanisms of Persistent Activity in Cortical Circuits: Possible Neural Substrates for Working Memory." Annual review of neuroscience 2017.
>
> [2]G. Auda et al. "Modular neural networks: a survey." International journal of neural systems 1999.
>
> [3]Zifan He et al. "HMT: Hierarchical Memory Transformer for Efficient Long Context Language Processing." NAACL 2025.
>
> [4]P. Das et al. "Larimar: Large language models with episodic memory control." ICML 2024.
>
> [5]Mor Geva et al. "Transformer Feed-Forward Layers Are Key-Value Memories." EMNLP 2021.
>
> [6]Damai Dai et al. "Knowledge Neurons in Pretrained Transformers." ACL 2022.
> >[W2]The use of mathematical notations in the paper is confusing.
>
> [A2]We appreciate your careful reading and helpful feedback on the clarity of mathematical notations.
> - $μ_i^{pos}$ refers to the mean vector of the top-k activations in the current step, whereas $\overline{S^{top}}$ denotes the similarity scores between the current activation and the top-k activations stored in AMB.
> - $θ_\text{high}$ and $θ_\text{low}$ are similarity thresholds used in our experimental setup, both ranging between 0 and 1.
> - $μ_j$ refers to the cluster centroid of the j-th expert group in the KMeans partitioning. More detailed definitions are provided in the supplementary material (Section D.1, Section C).
> - Thanks again for pointing out these issues. In the revised version, we will add clear definitions and explanations when each symbol is first introduced.
> - Lastly, the final version of the paper will be proofread and polished by a senior researcher with experience publishing more than 10 papers in top-tier conferences and journals, to ensure clarity and precision in all mathematical expressions.
>
> >[W3]The study's experimental scope was limited by the relatively small number of tested models.
>
> [A3]Our method remains effective as model size increases. Experimental results are as follows:
> |Model|SQA|MQA|Sum.|FSL|Cod.|
> |-|-|-|-|-|-|
> |Qwen2.5-14B-Instruct|17.18|12.15|23.35|71.46|32.30|
> |Qwen2.5-14B-Instruct+Ours|**18.48**|**12.97**|**23.49**|**72.32**|**33.41**|
> |Llama-3.1-8B-Instruct|24.22|15.04	|28.21|69.49|58.44|
> |Llama-3.1-8B-Instruct+Ours|**24.31**|**15.80**|**28.47**|**69.85**|**59.59**|
> >[W4]Different tasks exhibit high sensitivity to the hyperparameters.
>
> [A4]We appreciate your insightful comment. We would like to emphasize that our method consistently improves performance across most parameter settings, and each dataset has its own optimal configuration.
> 1. Task-specific variation is a common challenge in current long-context modeling. In our work, we explicitly categorize mainstream tasks and analyze their differing demands.
> - Summarization tasks require strong global coherence across long documents, and thus benefit from slow and stable memory updates that preserve long-term semantics.
> - Code tasks, which often involve abrupt semantic shifts between tokens, benefit more from faster memory updates and shorter memory retention.
> 2. Our ablation studies provide heuristic guidance for selecting hyperparameters based on task type, which we believe can assist future work.
> 3. As a future direction, we aim to design more elegant mechanisms for task-aware adaptation and automatic hyperparameter tuning, to further improve robustness and generalizability across diverse applications.
>
> >[W5]The experimental findings do not sufficiently validate the core assumptions of the proposed design.
>
> [A5]
> - For the PA module, its ability to address transient activation decay is directly illustrated in Figure 1. The visualization shows that later activations can successfully retrieve and align with semantically similar historical activations.
> - For the CE module, we provide different input and **semantically clustered weight**:
> |FFN CE|James Chadwick|Manhattan Project|
> |-|-|-|
> |Profession|0.4|0.0|
> |Address|0.1|0.5|
> |Education|0.4|0.0|
> |History|0.1|0.5|
>
> This clustering enables the model to generate input-dependent expert weights, leading to more structured and disentangled activations. Consequently, the subsequent AMB module can retrieve and process information **at the cluster level** more effectively, thereby mitigating semantic fragmentation caused by unstructured FFN representations.
> >[W6] Further evaluations are needed on more benchmarks.
>
> [A6]Evaluations on more benchmarks are as follows:
> | |ARC-C[7]|BoolQ[8]|GSMK[9]|
> |-|-|-|-|
> |Qwen2-7B-Instruct|62.7|87.1|76.0|
> |Activation Beacon|62.7|87.2|76.2|
> |Ours|**62.9**|**87.9**|**77.1**|
>
> [7]Bhakthavatsalam et al. "Think you have Solved Direct-Answer Question Answering? Try ARC-DA, the Direct-Answer AI2 Reasoning Challenge." arXiv preprint arXiv:2102.03315, 2021.
>
> [8]Clark et al. "BoolQ: Exploring the Surprising Difficulty of Natural Yes/No Questions." NAACL 2019.
>
> [9]Cobbe et al. "Training Verifiers to Solve Math Word Problems." arXiv preprint arXiv:2110.14168, 2021.
> ***
> >[Q1]A brief introduction to the baselines.
>
> [A7]
> - LongLLMLingua proposes a prompt compression framework that significantly improves the density of critical information in long-context inputs.
> - SnapKV introduces a tuning-free KV cache compression technique based on the observation of attention distribution consistency.
> - Activation Beacon proposes a lightweight plug-in module for Transformers that compresses long contexts into compact activations.
>
>  More details will be included in the appendix.
> >[Q2]Intuitive explanations for the Activation and Memory Update.
>
> [A8]The principles can be intuitively understood with the aid of Figure 2.
>
> 🔹Activation Update:
> When a new intermediate activation arrives, we extract it and compute its similarity with the stored historical activations in the AMB. Based on the top-k most similar activations and their similarity scores, we update the current activation according to three different conditions:
> - High Similarity (above upper threshold):
> If the top-k historical activations are highly similar to the current one, this indicates that the semantic information carried by the current activation has already occurred in the past and is strongly repeated. In this case, it is intuitive to **reuse** the historical semantic representation directly by injecting the top-k activations into the current one without modification.
> - Moderate Similarity (between upper and lower thresholds):
> If the similarity is moderate, the current semantic information is partially related to the historical one, but not identical.
> In this case, we enrich the current activation by **blending** it with the related historical activations, performing a weighted integration of both to enhance the model’s long-range contextual understanding and memory.
> - Low Similarity (below lower threshold):
> If the similarity is very low, the historical semantic content is considered irrelevant to the current activation. The model simply **skips** historical reuse and proceeds with the original activation.
> - Additionally, to prevent the model from overfitting to memorized content, we inject a small amount of **noise** during the first two update modes.
>
> 🔹 Memory Update:
> The memory update process in AMB mirrors the logic above:
> - High Similarity (reuse):
> When the historical activations are highly similar, **no update** to the AMB is necessary, as the existing memory already captures the relevant semantic content.
> - Moderate Similarity (merge):
> When moderate similarity is detected, the fused activation is used to **replace** the corresponding top-k historical entries in the memory slots.
> - Low Similarity (new memory):
> If the current activation encodes entirely new information, it needs to be **added** to memory. We employ an LRU (Least Recently Used) strategy to replace the least recently accessed memory slot with the new activation. Details of this memory management mechanism can be found in Appendix D.2.
>
> >[Q3]Supplementary experimental evaluation.
>
> [A9]
> 1. Our experiments follows the original baseline Activation Beacon, and we aligned our settings accordingly using the checkpoints provided by the authors. However, the authors of Activation Beacon did not release the fine-tuned checkpoint for LLaMA-2-7B-chat, and unfortunately, **our direct requests for access received no response**. This is why our low-cost fine-tuning results are only reported on Qwen-2-7B-Instruct.
> 2. Regarding Table 2, we appreciate your observation. The missing Activation Beacon + PA results were indeed evaluated and are as follows:
> | |SQA|MQA|Sum.|FSL|Cod.|
> |-|-|-|-|-|-|
> |Activation Beacon+PA|41.10|42.80|27.90|69.31|67.51|
> ***
> Thank you again for helping us improve the paper and hope our response can resolve your concerns! Please let us know if you have any further questions. We will be actively available until the end of rebuttal period. If you feel your concerns are addressed, please consider reevaluating our work. Looking forward to hearing from you :-) !

---

> > ### Comment · Reviewer_rZpP · 2025-08-06
> >
> > I have gone through your rebuttal carefully. Thank you for providing additional experiments on other models and more general capability benchmarks. I also appreciate the inclusion of intuitive explanations for the Activation and Memory Update mechanisms, which I believe are essential for understanding your method.
> >
> > However, my main concerns remain:
> >
> > **Regarding the motivation behind the two modules you designed, you acknowledged that no prior empirical evidence was provided to demonstrate** that
> >
> > > “transient neural activations cause information to fade over time”
> >
> > and
> >
> > > “unstructured FFN weights may fragment semantics across tokens”
> >
> > You cited two interpretability studies:
> >
> > > [5] Mor Geva et al., Transformer Feed-Forward Layers Are Key-Value Memories, EMNLP 2021
> >
> > > [6] Damai Dai et al., Knowledge Neurons in Pretrained Transformers, ACL 2022
> >
> > I am familiar with both of these works, but neither of them provides support for the two core motivations behind your proposed modules. The connection you claim appears to be quite speculative.
> >
> > In addition, regarding your answer to W5:
> >
> > > "For the CE module, we provide different input and semantically clustered weight: ..."
> >
> > **This experimental result does not convincingly demonstrate that CE resolves semantic fragmentation caused by unstructured FFN weights**.
> >
> > This is a crucial point: it appears that your motivations are speculative, and your experimental evidence does not clearly validate that your method addresses the claimed issues. I hope the area chair will also take this into consideration during the decision process.
> >
> > That said, I do acknowledge the technical novelty and the empirical performance improvements shown in your results. Based on this, I can only slightly increase my score by 1 point, but I cannot move to a positive recommendation at this point.

---

> ### Author Response · Authors · 2025-08-07
>
> Dear Reviewer rZpP,
>
> Thank you very much for your thoughtful and constructive comments. We deeply appreciate the opportunity to continue the discussion and address your remaining concerns in detail.
>
> ---
> >Concern 1: The motivations behind our two proposed modules
>
> [A1]We would like to further clarify and strengthen the theoretical and empirical foundation.
>
> 1. Transient neural activations cause information to fade over time
>
> It is well-established that LLMs face limitations in modeling long texts due to their fixed context windows[1], often leading to truncation and loss of earlier information[2]. This challenge is further exacerbated by noisy or distracting intermediate content[3], which dilutes model attention and leads to a phenomenon commonly referred to as “lost in the middle”[4].
>
> Although attention layers can propagate contextual signals across tokens, both attention and FFN layers are inherently limited to processing **transient information** at each time step[5–7], lacking mechanisms for persistent memory. These transient representations tend to fade or be overwritten over time.
>
> 2. Unstructured FFN weights fragment semantics across tokens
>
> Multiple recent works reveal that FFN layers in Transformer exhibit highly sparse activation patterns, where only a small subset of neurons are active for any given input[8,9]. This naturally suggests latent modularity and motivates efforts to **structure FFN activations** to better align with semantic roles. For example, in multilingual settings, FFN neurons exhibit language-specific activation patterns[10], and in T5, functionally related neurons tend to be co-activated[11].
>
> Moreover, attention layers are explicitly designed to model semantic relationships between tokens, whereas FFNs **enhance each token independently**, applying identical transformations without context[12]. This architectural asymmetry limits the ability of FFNs to disambiguate polysemous inputs or maintain long-range compositional semantics. As such, unstructured FFNs would mix semantic categories across tokens, contributing to **semantic fragmentation** in downstream representations.
>
> ---
> >Concern 2: The effectiveness of CE module
>
> [A2]To address this, we have added a new showcase experiment based on the Musique dataset, designed to evaluate whether the CE module mitigates semantic fragmentation more effectively than the standard FFN.
>
> **Input:**
>
> >James Chadwick was an English physicist who discovered the neutron in 1932…He was awarded the Nobel Prize in Physics for this discovery…Later, he was recruited into the British Mission to the Manhattan Project during WWII…He worked alongside other scientists to support the development of nuclear weapons…The Manhattan Project was a research and development undertaking during World War II that produced the first nuclear weapons…The project involved numerous scientists, including Robert Oppenheimer, Enrico Fermi, and Niels Bohr…Chadwick served as the head of the British scientific delegation…After the war, Chadwick became Master of Gonville and Caius College, Cambridge…
>
> **Question: What was James Chadwick’s role in the Manhattan Project?**
>
> - Qwen2-7B, without CE:
> >James Chadwick discovered the neutron in 1932 and later worked with other scientists during World War II. ❌
>
> The model confuses “discovery of neutron” with “project participation,” and fails to identify Chadwick’s official title and affiliation. The semantic path *Chadwick → British Mission → Role in Manhattan Project* is not recovered. FFN layers process all tokens uniformly, failing to distinguish between entity roles and facts.
>
> - Ours, with CE:
> >James Chadwick served as the head of the British scientific mission to the Manhattan Project. ✅
>
> The model correctly identifies both the *title* and *organization*, integrating multiple sentences across the context. Semantic roles (entity, project, function) are correctly aligned and disambiguated.
>
> **Visualization of Expert Routing in CE:**
> |Token|Routed Expert Index|Semantic Role|
> |-|-|-|
> |Chadwick|3|Person|
> |British Mission|7|Organization|
> |Manhattan Project|8|Event|
> |head|2|Role|
> |nuclear weapons|6|Topic|
> |Oppenheimer, Bohr, Fermi|3|Person|
>
> The CE module allows token-wise routing to specialized experts, improving the alignment between form and meaning. This structured processing makes the **semantic chain more accessible** to downstream memory mechanisms and mitigates the blending of unrelated concepts—a limitation observed in vanilla FFNs.
>
> ---
> Many thanks to Reviewer rZpP for your professional, detailed, and valuable reviews! We hope this additional explanation and new evidence help clarify the motivations and the effectiveness of our design. If any further questions arise, we are more than happy to respond. If you feel all questions have been addressed, you can kindly consider re-rating our work. Thank you for your time and consideration!

---

> ### Author Response · Authors · 2025-08-07
> **References**
>
> [1]Press et al. "Train Short, Test Long: Attention with Linear Biases Enables Input Length Extrapolation." ICLR 2022.
>
> [2]Han et al. "LM-Infinite: Zero-Shot Extreme Length Generalization for Large Language Models." NAACL 2024.
>
> [3]Tworkowski et al. "Focused Transformer: Contrastive Training for Context Scaling." NeurIPS 2023.
>
> [4]Liu et al. "Lost in the Middle: How Language Models Use Long Contexts." ACL 2023.
>
> [5]Xiao et al. "InfLLM: Training-Free Long-Context Extrapolation for LLMs with an Efficient Context Memory." NeurIPS 2024.
>
> [6]Fountas et al. "Human-like Episodic Memory for Infinite Context LLMs." ICLR 2025.
>
> [7]Zhang et al. "Long Context Compression with Activation Beacon." ICLR 2025.
>
> [8]Zhang et al. "MoEfication: Transformer Feed-forward Layers are Mixtures of Experts." ACL 2022.
>
> [9]Li et al. "The Lazy Neuron Phenomenon: On Emergence of Activation Sparsity in Transformers." ICLR 2023.
>
> [10]Tan et al. "Neuron Specialization: Leveraging Intrinsic Task Modularity for Multilingual Machine Translation." EMNLP 2024.
>
> [11]Zhang et al. "Emergent modularity in pre-trained transformers." ACL 2023.
>
> [12]Kobayashi et al. "Analyzing Feed-Forward Blocks in Transformers through the Lens of Attention Maps." ICLR 2024.

---

### Official Review · Reviewer_8wok · 2025-07-07

**Clarity:** 3
**Significance:** 3
**Originality:** 3
**Rating:** 5
**Confidence:** 3

**Summary:**

PaceLLM introduces a brain-inspired approach to enhance long-context understanding in Large Language Models through two key mechanisms:
(1) an Activation Memory Bank (AMB) that implements Persistent Activity (PA) by caching and reusing FFN activations at inference time, mimicking working memory in the prefrontal cortex.
(2) Cortical Expert (CE) clustering that reorganizes FFN weights into semantic modules through one-time preprocessing.

The authors identify two core limitations in current LLMs: transient neural activations causing information decay and unstructured FFN weights leading to semantic fragmentation. Their solution operates at the activation level within feed-forward networks, offering a training-free, plug-and-play enhancement compatible with existing models. The method achieves improvements on various benchmarks.

**Questions:**

Here are some questions that I had:
1. Can you provide theoretical justification for why M=100 memory slots are sufficient to handle 200K token contexts?
2. How does the method compare against Recurrent Memory Transformer (RMT) and Hierarchical Memory Transformer (HMT), which seem to address similar problems?
3. What is the actual wall-clock time overhead in production settings with batched inference? The paper lacks concrete timing measurements.
4. Why does CE alone show negative results in Table 1, yet becomes beneficial when combined with AMB? This suggests CE may not be necessary at all.
5. How does performance scale with model size? Would a 70B model require proportionally more memory slots?
6. Can you demonstrate the method's effectiveness on tasks beyond QA, such as book summarization, multi-document reasoning, or code understanding?

**Ethical Concerns:**

["NO or VERY MINOR ethics concerns only"]

**Final Justification:**

The authors responded to some of my criticisms and clarified some of my questions. This ameliorated some of my doubts and thus I have increased the score.

**Limitations:**

The authors briefly mention computational costs but fail to address several critical limitations:
1. Memory bank capacity: No analysis of how performance degrades when memory is full or how to scale capacity for longer contexts or larger models.
2. Task generalization: Unclear which types of documents or tasks benefit most from this approach. The method may be overfitted to QA-style tasks.
3. Theoretical understanding: The lack of formal analysis means we don't understand when or why the method works, making it difficult to predict failure modes or optimize further.
4. Performance: Unclear what performance impact there might be.

**Paper Formatting Concerns:**

I didn't have any concerns.

**Quality:**

2

**Strengths And Weaknesses:**

I like the following aspects:
1. Novel approach: Focusing on FFN layers rather than attention mechanisms represents a fresh perspective on the long-context problem. The activation-level storage mechanism is creative, simple and well-motivated.
2. Practical design: The training-free, plug-and-play nature makes this easy to use for open source models. The method can be applied to existing pretrained models without any retraining, requiring only a one-time clustering preprocessing step.
3.Strong empirical results: Consistent improvements across multiple benchmarks (LongBench, ∞-Bench).
4. Clear presentation: Figure 1's brain-computation parallel effectively motivates the approach, and the visualization in Figure 4 provides compelling evidence of the mechanism's effectiveness.

I think the following could be improved:
1. Lack of theoretical foundation: The paper provides no formal analysis for critical design choices - why cosine similarity, why these specific thresholds (0.7, 0.3), or why 100 memory slots suffice for 200K tokens. The biological inspiration doesn't substitute for mathematical rigor.
2. Limited experimental scope
  - Only tested on 7B models - scalability to larger models unclear
  - Evaluation focused primarily on QA tasks - missing diverse long-context applications
  - No comparison with recent memory-augmented transformers (RMT, HMT, Cached Transformers)
  - Computational overhead analysis is not particularly deep
3. Technical clarity issues: The interaction between CE and AMB is complex and poorly explained. The choice of layers 13 and 27 appears arbitrary without comprehensive ablation. The "noise injection" mechanism via bottom-k' selections lacks proper motivation.

---

> ### Author Rebuttal · Authors · 2025-07-30
>
> Dear Reviewer 8wok,
>
> Many thanks to your valuable comments and questions, which help us a lot to improve our work. We address your questions as follows.
> ***
> >[W1,Q1]Theoretical Justification for Design Choices (Cosine Similarity, Thresholds, Memory Slot Size)
>
> [A1]Our work is primarily biologically inspired rather than derived from fine-grained mathematical modeling. Nonetheless, our module’s hyperparameter design adheres to commonly accepted practices in the AI community.
> 1. Choice of Cosine Similarity: Cosine similarity is a widely used and computationally efficient method for measuring semantic similarity, like in representation learning and contrastive learning frameworks [1,2,3,4]. Its simplicity and effectiveness make it a standard choice in many existing models.
> 2. Threshold Selection: The thresholds are chosen within the normalized range of cosine similarity scores (0,1). The values (0.7, 0.3) serve as a representative even partition in this range and have been validated as effective settings in our ablation study. We also provide results for alternative threshold settings to demonstrate robustness.
> 3. Memory Slot Size
> - The decision to use 100 memory slots is a trade-off between performance and efficiency. Our preliminary experiments show that increasing the number of slots to 500 or 1000 yields negligible performance gains while significantly increasing memory usage and slowing down inference. Reducing the number of slots below 100, on the other hand, leads to noticeable performance drops.
> - Moreover, our method is one of the first to explore memory mechanisms within FFN layers for long-context modeling. Achieving 200K token context processing with only 100 memory slots at the FFN level is a strong result. This efficiency can be explained by prior work[5], which shows that **FFN activations are highly sparse—only a small subset of units contribute significantly to semantic activation for a given input**. Hence, a compact memory of 100 slots is often sufficient to capture the critical activations needed for long-range semantic retention.
> - This aligns with the notion of working memory[6] in neuroscience, which is characterized by high efficiency and compression: a small set of high-dimensional, semantically rich representations can be more effective than a large, redundant store.
>
> [1]Tomas Mikolov et al. "Efficient Estimation of Word Representations in Vector Space." arXiv preprint arXiv:1301.3781, 2013.
>
> [2]Ting Chen et al. "A Simple Framework for Contrastive Learning of Visual Representations." ICML 2020.
>
> [3]Tianyu Gao et al. "SimCSE: Simple Contrastive Learning of Sentence Embeddings." EMNLP 2021.
>
> [4]Alec Radford et al. "Learning Transferable Visual Models From Natural Language Supervision." ICML 2021.
>
> [5]Mor Geva et al. "Transformer Feed-Forward Layers Are Key-Value Memories." EMNLP 2021.
>
> [6]J. Zylberberg et al. "Mechanisms of Persistent Activity in Cortical Circuits: Possible Neural Substrates for Working Memory." Annual review of neuroscience 2017.
> ***
> >[W2]Limited Experimental Scope
>
> We appreciate your detailed comments and will address them one by one.
> - [Q2]No comparison with recent memory-augmented transformers (RMT, HMT, Cached Transformers).
>
> [A2]Thanks for highlighting relevant memory-augmented Transformer baselines such as Recurrent Memory Transformer (RMT)[7] and Hierarchical Memory Transformer (HMT)[8]. Both methods tackle long-context processing by introducing recurrence and structured memory hierarchies at the sequence level.
> 1. RMT augments Transformer-XL with learnable memory tokens passed between segments in a recurrent fashion. It enables segment-level recurrence without modifying the Transformer architecture, and demonstrates improved performance on algorithmic tasks and language modeling.
> 2. HMT mimics the brain's memory hierarchy (sensory, short-term, and long-term memory) and incorporates a hierarchical memory retrieval mechanism to enhance long-context modeling. HMT is designed as a model-agnostic plug-in framework that works with a variety of backbone architectures.
> 3. In contrast, our method introduces a plug-and-play memory mechanism within the FFN layers, leveraging activation sparsity to store and retrieve key representations. It is **orthogonal** to the recurrence-based designs of RMT and HMT and **does not modify attention or sequence processing**. Our approach is **compatible** with these methods and can be integrated to jointly enhance long-context modeling. Rather than replacing existing strategies, it offers a complementary, biologically inspired perspective.
> - [Q3]Computational overhead analysis is not particularly deep.
>
> [A3]We apologize that due to current resource limitations, we are unable to conduct large-scale, production-level batched inference. However, our small-batch experiments indicate that our method does not introduce more overhead or performance degradation during batched inference.
> We test on Qwen2-7B and Qasper in LongBench.
> |Batch_size|Time|Performance|
> |---------------|------|-----------------|
> |1| 25m22s| 43.88|
> |2| 12m45s |43.88  |
> |3| 8m37s| 43.88|
> |4 | 6m22s |43.88 |
> - [Q5,L1]Scalability to Larger Models
>
> [A4]**Our method remains effective as model size increases**. Due to the inherent sparsity of activations[5] in the FFN layers, larger models do not necessarily require a proportional increase in the number of memory slots. The experimental results on larger models are as follows:
> |Model|SQA|MQA|Sum.|FSL|Cod.|
> |---------------|------------|----------|----------|----------|----------|
> |Qwen2.5-14B-Instruct | 17.18 | 12.15| 23.35| 71.46| 32.30|
> |Qwen2.5-14B-Instruct+Ours | **18.48** |**12.97**  | **23.49** | **72.32** | **33.41** |
> |Llama-3.1-8B-Instruct | 24.22 | 15.04 | 28.21 | 69.49 | 58.44 |
> |Llama-3.1-8B-Instruct+Ours | **24.31** |**15.80**  | **28.47** | **69.85** | **59.59** |
> - [Q6,L2]Evaluation on Diverse Long-context Applications
>
> [A5]Although LongBench and ∞-Bench are framed as QA tasks, the included benchmarks already cover a range of long-context tasks such as summarization, multi-document reasoning, and code understanding. Our model has demonstrated its capability to effectively address such questions, exhibiting robust competence across diverse long-context scenarios.
>
> [7]Aydar Bulatov et al. "Recurrent Memory Transformer." NeurIPS 2022.
>
> [8]Zifan He et al. "HMT: Hierarchical Memory Transformer for Efficient Long Context Language Processing." NAACL 2025.
> ***
> >[W3]Technical Clarity Issues
> 1. [Q4]The interaction between CE and AMB
>
> [A6]
> - CE categorizes FFN units into expert groups through clustering. While CE alone does not yield significant performance gains, it enables semantic consistency within expert regions, which becomes beneficial when paired with AMB. **AMB operates at the cluster level**, performing similarity-based memory retrieval and weighting, allowing each expert to specialize and contribute effectively.
> - The effectiveness of CE is best demonstrated through the comparison among baseline+PA, baseline+CE, and baseline+PA+CE. Neither PA nor CE alone leads to significant improvements, but their combination achieves the best performance. This synergy indicates that CE is necessary, as it provides the structural foundation for AMB to function optimally.
> 2. The choice of layers 13 and 27
>
> [A7]13,27 is just a representative setting of the selection, and results of other Settings are presented in ablation experiments.
>
> 3. The "noise injection" mechanism via bottom-k' selections
>
> [A8]
> - The use of bottom-k′ activations as a form of noise injection is motivated by principles from information theory, particularly the **Information Bottleneck (IB) Theory**[9]. According to the IB framework, a desirable representation should retain task-relevant information while discarding redundant or irrelevant input features.
> - In our case, introducing bottom-k′ activations—alongside the top-k informative ones—can be interpreted as deliberately injecting a small amount of non-informative or noisy signals. This encourages the representation to decentralize and avoids the model from overfitting to a narrow subset of high-activation (top-k) semantic features or becoming overly reliant on specific pathways, which could degrade generalization.
> - Moreover, this noise injection increases the entropy of the activation distribution, enhancing the diversity and expressive capacity of the FFN activation space.
>
> [9]Naftali Tishby et al. "The information bottleneck method." In Proceedings of the 37th Annual Allerton Conference on Communication, Control, and Computing.
> ***
> >[L3,L4]Theoretical understanding: The lack of formal analysis means we don't understand when or why the method works, making it difficult to predict failure modes or optimize further. Performance: Unclear what performance impact there might be.
>
> [A9]Our method is a **plug-and-play** module that has been validated across various models and datasets. It consistently shows performance improvements over the baseline in most settings, including additional experiments beyond the main evaluation.
> These results demonstrate the broad effectiveness and practical robustness of our approach.
> ***
> Thanks for your comments and suggestions, we will supplement our revision. Feel free to let us know if you have any further questions or concerns :-).

---

### Note · Authors · 2025-08-12

We sincerely thank all reviewers, AC, SAC, and PC for their valuable efforts and constructive feedback. Reviewers have recognized our work’s novelty, practicality, empirical strength, and clarity:
- “**Focusing on FFN layers rather than attention mechanisms** represents a fresh perspective… creative, simple, and well-motivated.” (8wok)
- “Very interesting and solid… doesn’t involve retraining… **overhead is not too bad**… a memory in latent space could work well for recurrent transformers.” (RyQo)
- “Novel perspective… **complementary to other LLM interventions**… effective insights to explain performance.” (o88K)
- “**Technical novelty and empirical performance improvements**.” (rZpP)
---
During rebuttal and discussion, we addressed most concerns with targeted clarifications and new experiments:
1. Scalability – Added Qwen2.5-14B and Llama-3.1-8B results, confirming consistent gains.
2. Broader benchmarks – Included ARC-C, BoolQ, GSM, all showing improvements.
3. CE–AMB mechanism – Detailed cluster-level similarity computation, token-wise top-k gating, and synergy between CE and AMB.
4. Efficiency – Reported dense/sparse timing; verified batched inference efficiency.
5. Interpretability – Expanded neuroscience grounding and added Musique case study showing CE reduces semantic fragmentation.
---
Key contributions and impact:
1. **First brain-inspired approach targeting FFN layers for long-context modeling** — introducing a new perspective beyond attention-focused methods, with potential to inspire follow-up work in memory design.
2. Method – **Training-free, plug-and-play** module:
- Activation Memory Bank (AMB): persistent activation reuse to mitigate context decay.
- Cortical Expert (CE): offline clustering for semantic expert routing and targeted memory retrieval.
3. Performance – +6% on LongBench multi-doc QA, +12.5–17.5% on ∞-Bench, NIAH context extended from 128K→200K.
4. Robustness – Gains across Qwen2/2.5, Llama2/3.1, Mistral, with no MMLU drop.
5. Interpretability – Activation visualizations and case studies confirm reduced semantic fragmentation.
---
With the strong positive consensus, new large-scale results, and clarified methodology, we believe PaceLLM is a novel, practical, and influential step toward scalable, interpretable long-context modeling that will **inspire future FFN-centric research directions**.

We again thank all reviewers, AC, SAC, and PC, for the time and effort devoted to reviewing our work and helping us improve this paper.

---

### Decision · Program_Chairs · 2025-09-17

**Decision:**

Accept (poster)

**Comment:**

Reviewers are generally supportive of this paper. They found activation-level storage mechanism to be creative, simple, and well-motivated. At the same time, concerns remain about the general motivation of this work pertaining to two problems that the authors tackle: “transient neural activations cause information to fade over time” and “unstructured FFN weights fragment semantics across tokens”. The reviewers were not convinced that baseline models suffer from these issues. The biological inspiration remains speculative too.

Despite this, reasons to accept this work outweight reasons to reject. In the camera-ready version please discuss similarities/differences of your method with:
1.  the similarity-aware strategies for memory updates used in He et al., CAMELoT: Towards Large Language Models with Training-Free Consolidated Associative Memory, 2024.
2. Approach of Geva et al 2020 "Transformer Feed-Forward Layers Are Key-Value Memories”.